# Latent Variable Estimation in Bayesian Black-Litterman Models

**Thomas Y.L. Lin** [1]  **Jerry Yao-Chieh Hu** [2][3]  **Paul W. Chiou** [4]  **Peter Lin** [5][6]

## Abstract

We revisit the Bayesian Black–Litterman (BL) portfolio model and remove its reliance on subjective investor views. Classical BL requires an investor "view": a forecast vector $q$ and its uncertainty matrix $\Omega$ that describe how much a chosen portfolio should outperform the market. Our key idea is to treat $(q, \Omega)$ as latent variables and learn them from market data within a single Bayesian network. Consequently, the resulting posterior estimation admits closed-form expression, enabling fast inference and stable portfolio weights. Building on these, we propose two mechanisms to capture how features interact with returns: shared-latent parametrization and feature-influenced views; both recover classical BL and Markowitz portfolios as special cases. Empirically, on 30-year Dow-Jones and 20-year sector-ETF data, we improve Sharpe ratios by 50% and cut turnover by 55% relative to Markowitz and the index baselines. This work turns BL into a fully data-driven, view-free, and coherent Bayesian framework for portfolio optimization.

## 1 Introduction

We propose a Bayesian reformulation of the Black-Litterman model for portfolio optimization. Our motivation comes from the early works of the model (Black & Litterman, 1992; Lee, 2000; Salomons, 2007; Idzorek, 2007) where human experts are required to specify the value of investor views and corresponding uncertainties $(q, \Omega)$. For example, the $i$-th views "the 2nd asset will outperform the 1st asset by $9 \pm 3\%$" is encoded as $\{P_i = [-1, 1, ..., 0]; q_i = 0.09; \Omega_{ii} = 0.03^2\}$. Across decades, this heuristic framework attracts many research (Beach & Orlov, 2007; Palomba, 2008; Duqi et al., 2014; Silva et al., 2017; Deng, 2018; Kara et al., 2019; Kolm & Ritter, 2021) working on this estimation. Among them, one common approach is to take asset features and generate $(q, \Omega)$ by external models. However, relying on external estimators in these methods leads to incoherent parameter learning and error propagation across the separate models.

In this work, we offer a new approach. Our reformulation recasts the Black-Litterman model as a Bayesian network to integrate the features. Under different scenarios, we identify two potential effects caused by the features and accordingly present this network as specific models. Under the scenarios without subjective investor views, this network treats $q$ and $\Omega$ as latent — rather than externally estimated — parameters, and thereby estimates posterior distribution over both asset returns $r$ and their parameters $\theta$ directly from data, i.e., features. In summary, our approach provides a unification of feature integration and parameter inference within a single framework, ensuring coherent estimations and mitigating error propagation.

**Contributions.** Our contributions include:

- **Eliminating Subjective Human Input.** We introduce a Bayesian network formulation of the Black-Litterman model, treating $(q, \Omega)$ as latent variables. This enables direct estimation from feature data, bypassing heuristic human inputs and potential bias while subsuming the classical Black-Litterman model as a special case.

- **Unified Feature Integration and Parameter Inference.** Unlike prior works, our approach avoids error propagation from external estimators by unifying feature integration and parameter inference into a single framework.

- **Empirical Outperformance.** Our model achieves a 49.8% mean improvement in Sharpe ratios over the Markowitz model (0.66–0.87 vs. 0.35–0.62) and market indices (S&P 500, DJIA) on 20-year and 30-year datasets, respectively. It achieves a 55.1% reduction in turnover rates while showing robustness to hyperparameters.

---

*Equal contribution  [1]Department of Physics, National Taiwan University, Taipei, Taiwan [2]Department of Computer Science, Northwestern University, Evanston, IL, USA [3]Center for Foundation Models and Generative AI, Northwestern University, Evanston, IL, USA [4]D'Amore-McKim School of Business, Northeastern University, Boston, MA, USA [5]Gamma Paradigm Capital, New York, NY, USA [6]Whiting School of Engineering, Johns Hopkins University, Baltimore, MD, USA. Correspondence to: Thomas Y.L. Lin <b12202026@ntu.edu.tw>, Jerry Yao-Chieh Hu <jhu@u.northwestern.edu>, Paul W. Chiou <w.chiou@northeastern.edu>, Peter Lin <peter.lin@jhu.edu>.

*Proceedings of the 42$^{nd}$ International Conference on Machine Learning*, Vancouver, Canada. PMLR 267, 2025. Copyright 2025 by the author(s).

**Organization.** Section 2 includes preliminaries. Section 3 introduces our models and their theoretical analysis. Section 4 presents empirical studies to backup our work. Appendix A offers a practical guide for our models. We defer conclusions and related works to Section 5 and Appendix B.

## 2 Preliminaries

Consider $m$ assets, and let $r \in \mathbb{R}^m$ be the returns of the $m$ assets. Consider $k$ sets of specified portfolio weights on the $m$ assets, and encode each weight into each row of a portfolio weight matrix $P \in \mathbb{R}^{k \times m}$. Let $q \in \mathbb{R}^k$ be the investor views on the $k$ specified portfolio returns and encode the variance of each view into diagonal elements of a diagonal uncertainty matrix $\Omega \in \mathbb{R}^{k \times k}$. Larger $\Omega_{ii}$ implies greater uncertainty in $(P\mathbb{E}[r])_i$ and $\Omega_{ii} = 0$ implies absolute certainty.

Markowitz (1952) introduces the theory of portfolio optimization, suggesting a suitable portfolio weight is the optimal trade-off between the mean and variance of the portfolio. To be concrete, we provide a formal definition below.

**Definition 2.1** (Unconstrained Risk-Adjusted Mean-Variance Optimization)**.** Let $r \in \mathbb{R}^m$ be the returns of the $m$ assets, $\widetilde{r}$ denote the unobserved (or future) asset returns, and $\delta \in [0, \infty]$ be a risk-adjusted coefficient. The optimization goal is to find $w \in \mathbb{R}^m$ maximizing the objective function:

$$\max_w \left\{ w^T \mathbb{E}[\widetilde{r}] - \frac{\delta}{2} w^T \text{Cov}[\widetilde{r}]w \right\}.$$

One major challenge of this framework is the reliance on estimating $\widetilde{r}$:

**Problem 1** (Predictive Estimation of Unobserved Asset Returns)**.** Let $r \in \mathbb{R}^m$ represent the returns of the $m$ assets and $\widetilde{r}$ denote the unobserved (or future) asset returns. Precisely, given observed data $D$, the goal is to estimate unobserved asset returns $\widetilde{r} \sim p(r|D)$.

In this work, we refer to methods addressing such estimation challenge (Problem 1) as portfolio models or simply portfolios. Following the predictive estimation of $\widetilde{r}$, we apply the mean-variance optimization framework to obtain a decision vector $w$, referred to as portfolio weights.

A basic approach, termed the traditional Markowitz model (Markowitz, 1952), involves predicting the expected returns and the covariance matrix of asset returns directly from historical data using the sample mean and sample covariance. This method relies on the assumption that historical estimates are accurate representations of future parameters. However, in practice, estimation errors in the expected returns and covariance matrix lead to extreme and highly sensitive portfolio weights (Michaud, 1989; DeMiguel et al., 2009). To mitigate this issue, the Black-Litterman model integrates the market equilibrium with investor views by Black-Litterman formula, thereby producing more stable

and diversified portfolio weights (Black & Litterman, 1992). The following context elaborates on this model in detail.

**Black-Litterman.** Black-Litterman (BL) model outputs a posterior of the asset returns mean $\mathbb{E}[r]$, termed Black-Litterman formula, by Bayes' theorems, taking investor views and market equilibrium price as input. Upon this, the model offers a predictive estimate $\widetilde{r}$ on asset returns:

**Theorem 2.1** (Black-Litterman (BL) Formula and Predictive Estimation, Theorem 1 of (Satchell & Scowcroft, 2007))**.** Let $r \in \mathbb{R}^m$ be the vector of asset returns with covariance $\Sigma := \text{Cov}[r]$. Let $P \in \mathbb{R}^{k \times m}$ be the portfolio weight matrix for $k$ specified portfolios, and $(q, \Omega) \in \mathbb{R}^k \times \mathbb{R}^{k \times k}$ represent investor views and their uncertainty. Let $\Pi \in \mathbb{R}^m$ represent the market equilibrium price and $\tau > 0$ be a scaling factor. Assume a prior $P\,\mathbb{E}[r] \sim N(q, \Omega)$ and a likelihood $\Pi \mid \mathbb{E}[r] \sim N\big(\mathbb{E}[r], \tau\,\Sigma\big)$, then the posterior mean of $r$ given $\Pi$ is

$$\mathbb{E}[r \mid \Pi] \sim N\Big(G_\tau^{-1}\big[(\tau\Sigma)^{-1}\Pi + P^\top \Omega^{-1} q\big], G_\tau^{-1}\Big),$$

where $G_\tau := (\tau\,\Sigma)^{-1} + P^\top \Omega^{-1} P$. Moreover, the predictive distribution $\widetilde{r} := r|\Pi$ is

$$\widetilde{r} \sim N\Big(G_\tau^{-1}\big[(\tau\Sigma)^{-1}\Pi + P^\top \Omega^{-1} q\big], \Sigma + G_\tau^{-1}\Big).$$

The Black-Litterman formula (Theorem 2.1) is a well-known result of the Black-Litterman model. However, the derivation lacks explanations of the assumptions used. Most early works, including the original paper (Black & Litterman, 1992), provide heuristic derivation, while many (Lee, 2000; Salomons, 2007; Idzorek, 2007) share different underlying assumptions. This inconsistency leads to confusion for both the analysis of the model and a rigorous interpretation with Bayesian statistics. To solve the issues, the Black-Litterman-Bayes (BLB) model (Kolm & Ritter, 2017) provides a reformulation of the Black-Litterman model.

**Black-Litterman-Bayes (BLB).** Kolm & Ritter (2017) introduce the Black-Litterman-Bayes model to perform Bayesian inference on $\theta$, treating the market equilibrium as prior and the investor views as likelihood:

**Definition 2.2** (BLB Model $(\theta, r, q, \Omega)$, Modified from Definition 1 of (Kolm & Ritter, 2017))**.** Let $r \in \mathbb{R}^m$ be the returns of the $m$ assets, parametrized by $\theta$, with $r \sim p(r|\theta)$. Let $q \in \mathbb{R}^k$ represent the views on the returns of the $k$ specified portfolio and $\Omega \in \mathbb{R}^{k \times k}$ be the uncertainty matrix. The Black-Litterman-Bayes (BLB) model is a portfolio model composed of three fundamental density functions:

1. Parametrized Asset Returns: $p(r|\theta)$, the distribution of asset returns given the parameter.

2. Prior: $\pi(\theta)$, representing market equilibrium.

3. Likelihood: $L(\theta|q, \Omega) := p(q, \Omega|\theta)$, capturing the relationship between the parameter and the investor views.

Appendix C.1 details modelings of the prior and likelihood.

To recap, the Black-Litterman-Bayes model is a portfolio model aiming to address the estimation challenge of asset returns (Problem 1). It solves the problem by using Bayesian inference to obtain the posterior of the model parameter $p(\theta|q, \Omega)$, and subsequently produce the predictive estimation of unobserved asset returns $\widetilde{r}$. Following the estimation, we apply the mean-variance optimization framework (Definition 2.1) to obtain the portfolio weights $w_{\mathrm{BLB}}$.

The intuition of the model is to simulate the dynamics of how the market adapts to new information after observing it. Here, the market equilibrium represents the initial state, and the investor views approximates the unobserved information. When there is no investor views, the model remains in the initial state of market equilibrium.

Here we show the Black-Litterman formula (Theorem 2.1) is the posterior estimation on $\theta$ of the Black-Litterman-Bayes model under Assumptions C.1 and C.2:

**Lemma 2.1** (Estimations by BLB Model). Let the market capitalization weight on $m$ assets be $w_{\mathrm{cap}} \in \mathbb{R}^m$ and $\delta \in [0, \infty]$ be a risk-adjusted coefficient. Let $P \in \mathbb{R}^{k \times m}$ be the portfolio weight matrix for $k$ specified portfolios. Given a BLB model $(\theta, r, q, \Omega)$ (Definition 2.2), assume

$$r \sim N(\theta, \Sigma), \tag{2.1}$$

$$\theta \sim N(\theta_0, \Sigma_0), \tag{2.2}$$

$$P\theta = q + \epsilon, \quad \epsilon \sim N(0, \Omega), \tag{2.3}$$

where $\Sigma, \Sigma_0 \in \mathbb{R}^{m \times m}$ are given intrinsic and prior covariance. The posterior mean is

$$p(\theta|q, \Omega) = N\left(\theta; G^{-1}\left(\Sigma_0^{-1}\theta_0 + P^{\mathsf{T}}\Omega^{-1}q\right), G^{-1}\right), \tag{2.4}$$

where $G := \Sigma_0^{-1} + P^{\mathsf{T}}\Omega^{-1}P$. The predictive estimation of unobserved asset returns $\widetilde{r} := r|q, \Omega$ is

$$\widetilde{r} \sim N\left(\widetilde{r}; G^{-1}\left[\delta(\Sigma_0^{-1}\Sigma + I)w_{\mathrm{cap}} + P^{\mathsf{T}}\Omega^{-1}q\right], \Sigma + G^{-1}\right). \tag{2.5}$$

*Proof.* See Appendix E.1 for a detailed proof. □

Lemma 2.1 provides a solution to Problem 1. With the predictive estimation of asset returns $\widetilde{r}$, the mean-variance optimization framework (Definition 2.1) determines the portfolio weights $w_{\mathrm{BLB}}$. However, it relies on the subjective investor views and corresponding uncertainty $(q, \Omega)$[1].

---

[1] Besides providing the Bayesian inference formulation of the Black-Litterman model (Definition 2.2), the original work (Kolm & Ritter, 2017) and its follow-up work (Kolm & Ritter, 2021) consider external data, specifically factors in Arbitrage Pricing Theory (APT) model. Yet, these works focus on how their Black-Litterman-Bayes (BLB) approach applies to the APT model and do not address the issues of subjective investor views.

To address this issue, in this work, we propose a Bayesian reformulation of the Black-Litterman model without the need for subjective $(q, \Omega)$ from humans.

## 3 Method

In this work, we recast the Black-Litterman model as Bayesian networks for principled estimation of both investor views and asset returns, eliminating the need for subjective inputs. This Bayesian formulation serves as a conceptual baseline for subsequent portfolio model specifications.

In Section 3.1, we introduce the Bayesian Black-Litterman network, which underpins the Black-Litterman-Bayes model (Kolm & Ritter, 2017). Building on this, Section 3.2 extends the network to incorporate external features, yielding the feature-integrated Black-Litterman network.

We then examine two scenarios:

- In Section 3.3, where investor views are observed, we illustrate the corresponding network (Figure 2) and define the Mixed-effect Black-Litterman (M-BL) model (Definition 3.3).

- In Section 3.4, where no subjective views are given, we present two alternative probabilistic graphical models (Figure 3) and define the Shared-Latent-Parametrization Black-Litterman (SLP-BL) and Feature-Influenced-Views Black-Litterman (FIV-BL) models (Definitions 3.4 and 3.6).

### 3.1 Bayesian Black-Litterman Network

We introduce a Black-Litterman network $(\theta, r, q, \Omega)$ with the two causal relationships. First, the asset returns $r$ are realizations of the process governed by its parameter $\theta$. Second, the investor views $q$ are formed based on $\theta$ with an associated error term $\epsilon \sim N(0, \Omega)$. We visualize this conceptual network in Figure 1.

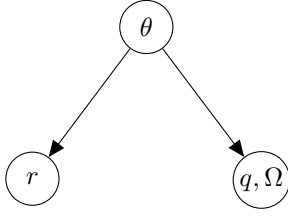

Figure 1: Black-Litterman network $(\theta, r, q, \Omega)$.

### 3.2 Bayesian Feature-Integrated Black-Litterman Network

Building upon the Black-Litterman Network, we introduce a feature-integrated Black-Litterman network as a Black-Litterman network that integrates features $F$ and their effects. Specifically, features $F$ exert two causal effects:

- **Effect 1**: Features $F$ are extracted from the parameter $\theta$.

- **Effect 2**: Features $F$ influence the formation of views $q$.

We quantitatively specify Effect 1 and 2 in Sections 3.3 and 3.4.

### 3.3   General Scenario: Features with Observed Views

In this section, we discuss the general scenario where views are observed (Problem 2) by specifying the two causal effects introduced in Section 3.2. Incorporating these effects into the network, we showcase it in Figure 2 and define the Mixed-effect Black-Litterman (M-BL) model (Definition 3.3) based on it. Then, we estimate posterior distribution over both asset returns $r$ and their parameter $\theta$ (Corollary 3.1.1 and Theorem 3.1).

Consider the following problem:

**Problem 2** (Feature-and-Views Hybrid Predictive Estimation). Let $r \in \mathbb{R}^m$ represent the returns of the $m$ assets and $\widetilde{r}$ denote the unobserved (or future) asset returns. Let $q \in \mathbb{R}^k$ represent the views on returns of the $k$ portfolios. Let $f_i \in \mathbb{R}^d$ represent the features on the $i$-th asset and $F \in \mathbb{R}^{m \times dm}$ be a block-diagonal matrix defined as
$$F := \operatorname{diag}(f_1^\mathsf{T}, f_2^\mathsf{T}, \ldots, f_m^\mathsf{T}).$$
Given $D := (q, \Omega, F, \Omega^F)$ where $(q, \Omega, F)$ are estimated from observations of asset returns, views, and features $\{(r_l, q_l, F_l)\}_{l=1}^n$ and $\Omega^F$ is the homoscedastic error matrix corresponding to the observations $\{F_l\}_{l=1}^n$, the goal is to estimate unobserved asset returns $\widetilde{r} \sim p(r|D)$.

We aim to solve Problem 2 by feature-integrated Black-Litterman network, incorporating a mix of two causal effects of features in Section 3.2. Specifically,

- **Effect 1**: Features $F$ are extracted from the parameter $\theta$. Consequently, the features $F$, along with their error term $\epsilon^F \sim N(0, \Omega^F)$, share the common parameter $\theta$ with asset returns $r$ and investor views $q$.

- **Effect 2**: Features $F$ influence the formation of views $q$. Consequently, the features $F$ and the parameter $\theta$ jointly determine the views $q$ with an uncertain $\epsilon \sim N(0, \Omega)$.

We visualize the network in Figure 2 under this general scenario.

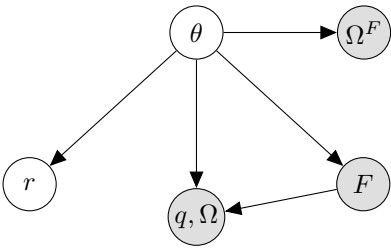

Figure 2: Feature-Integrated BL network with features and observed views $(\theta, r, q, \Omega, F, \Omega^F)$.

To capture Effect 1, we define a $\theta \leftrightarrow F$ relationship:

**Definition 3.1** ($\theta \leftrightarrow F$ Linear Model). Given features $F \in \mathbb{R}^{m \times dm}$, let $r \in \mathbb{R}^m$ be the returns of the $m$ assets, parametrized by $\theta$, with $r \sim p(r|\theta)$. Define regression intercept vector $\alpha^F \in \mathbb{R}^m$, regression coefficient vector $\beta^F \in \mathbb{R}^{dm}$, random error $\epsilon^F \in \mathbb{R}^m$, and error matrix $\Omega^F \in \mathbb{R}^{m \times m}$ such that:
$$\theta = \alpha^F + F\beta^F + \epsilon^F, \quad \epsilon^F \sim N(0, \Omega^F).$$

To capture Effect 2, we define a $q \leftrightarrow F \leftrightarrow \theta$ relationship:

**Definition 3.2** ($q \leftrightarrow F \leftrightarrow \theta$ Linear Model). Given features $F \in \mathbb{R}^{m \times dm}$ and portfolio weight matrix for $k$ specified portfolios $P \in \mathbb{R}^{k \times m}$, let $r \in \mathbb{R}^m$ be the returns of the $m$ assets, parametrized by $\theta$, with $r \sim p(r|\theta)$ and $q \in \mathbb{R}^k$ be the views on returns of the $k$ portfolios. Define regression intercept vector $\alpha \in \mathbb{R}^m$, regression coefficient vector $\beta \in \mathbb{R}^{dm}$, scale constant $\gamma \in \mathbb{R}$, random error $\epsilon \in \mathbb{R}^m$, and uncertainty matrix $\Omega \in \mathbb{R}^{m \times m}$ such that:
$$q + \epsilon = P(\alpha + F\beta + \gamma\theta), \quad \epsilon \sim N(0, \Omega). \tag{3.1}$$

**Remark 3.1** (Rationale). (3.1) extends the classical noisy views model $q + \epsilon = P\theta$ (Black & Litterman, 1992) to incorporate the features $F$, where the LHS remains the $k$-dimensional noisy views and the RHS generalizes the classical $P\theta$ term by introducing a feature-driven term $\alpha + F\beta$, and a scaled parameter $\gamma\theta$. It recovers the classical model when $\alpha = 0$, $\beta = 0$, and $\gamma = 1$.

By mixing two effects of the features characterized by the two linear models (Definitions 3.1 and 3.2), we showcase the feature-integrated Black-Litterman network as the following Mixed-effect Black-Litterman (M-BL) model:

**Definition 3.3** (Mixed-effect Black-Litterman (M-BL) Model $(\theta, r, q, \Omega, F, \Omega^F)$). Let $r \in \mathbb{R}^m$ be the returns of the $m$ assets, parametrized by $\theta$, with $r \sim p(r|\theta)$. Let $q \in \mathbb{R}^k$ represent the views on the returns of the $k$ specified portfolio and $\Omega \in \mathbb{R}^{k \times k}$ be the uncertainty matrix. Let $F \in \mathbb{R}^{m \times dm}$ be the features of the $m$ assets and error matrix $\Omega^F \in \mathbb{R}^{m \times m}$ The M-BL model is a portfolio model composed of four fundamental density functions:

1. Parametrized Asset Returns: $p(r|\theta)$, the distribution of asset returns given the parameter.

2. Prior: $\pi(\theta)$, representing market equilibrium.

3. Likelihood of Features: $L(\theta|F, \Omega^F) := p(F, \Omega^F|\theta)$, the $\theta \leftrightarrow F$ relationship (Definition 3.1).

4. Observation Likelihood: $L(\theta, F|q, \Omega) := p(q, \Omega|\theta, F)$, the $q \leftrightarrow F \leftrightarrow \theta$ relationship (Definition 3.2).

We show the posterior estimation on $\theta$ of the M-BL model:

**Theorem 3.1** (Parameter Estimation of the M-BL Model). Given a M-BL model $(\theta, r, q, \Omega, F, \Omega^F)$ (Definition 3.3) and regression parameters $(\alpha^F, \beta^F \alpha, \beta, \gamma) \in \mathbb{R}^m \times \mathbb{R}^{dm} \times$

$\mathbb{R}^m \times \mathbb{R}^{dm} \times \mathbb{R}$, assume

$$\theta \sim N(\theta_0, \Sigma_0), \tag{3.2}$$

$$\theta = \alpha^F + F\beta^F + \epsilon^F, \quad \epsilon^F \sim N(0, \Omega^F), \tag{3.3}$$

$$q + \epsilon = P(\alpha + F\beta + \gamma\theta), \quad \epsilon \sim N(0, \Omega). \tag{3.4}$$

Define $G^M := \Sigma_0^{-1} + (\Omega^F)^{-1} + \gamma^2 P^\mathsf{T} \Omega^{-1} P$. The posterior mean is

$$p(\theta|q, \Omega, F, \Omega^F) = N(\theta; \mu_{\theta|q,\Omega,F,\Omega^F}, (G^M)^{-1}),$$

where

$$\mu_{\theta|q,\Omega,F,\Omega^F} = (G^M)^{-1} \left[ \Sigma_0^{-1}\theta_0 + (\Omega^F)^{-1}(\alpha^F + F\beta^F) \right.$$
$$\left. + \gamma P^\mathsf{T} \Omega^{-1}(q - P\alpha - PF\beta) \right].$$

*Proof.* See Appendix E.2 for a detailed proof. □

The posterior estimation of parameter $\theta$ enables a predictive estimation on $\widetilde{r}$:

**Corollary 3.1.1** (Predictive Estimation by the M-BL Model). Define $G^M := \Sigma_0^{-1} + (\Omega^F)^{-1} + P^\mathsf{T} \Omega^{-1} P$. Assume $r \sim N(\theta, \Sigma)$. Then, under Theorem 3.1, M-BL model gives the predictive estimation of unobserved asset returns $\widetilde{r} := r \mid q, \Omega, F, \Omega^F$ as

$$\widetilde{r} \sim N(\theta; \mu_{\theta|q,\Omega,F,\Omega^F}, \Sigma + (G^M)^{-1}),$$

where

$$\mu_{\theta|q,\Omega,F,\Omega^F} = (G^M)^{-1} \left[ \Sigma_0^{-1}\theta_0 + (\Omega^F)^{-1}(\alpha^F + F\beta^F) \right.$$
$$\left. + \gamma P^\mathsf{T} \Omega^{-1}(q - P\alpha - PF\beta) \right].$$

**Remark 3.2** (Classical Black-Litterman Recovery). M-BL model recovers classical Black-Litterman model when:

1. *Features becomes uninformative, i.e., uncertainty approaches infinity*: $\Omega^F \to \infty$, equivalently $(\Omega^F)^{-1} \to 0$, so the error matrix $(\Omega^F)^{-1}$ disappears from $G^M$.

2. *The $q \leftrightarrow F \leftrightarrow \theta$ linear model (Definition 3.2) reduces to the classical noisy views model (Black & Litterman, 1992)*: $(\alpha, \beta, \gamma) = (0, 0, 1)$, so the residual $(q - P\alpha - PF\beta) \to q$.

Under these conditions, we have

$$G^M \to \Sigma_0^{-1} + P^\mathsf{T} \Omega^{-1} P = G,$$

and thus, we recover (2.5):

$$r|q, \Omega \sim N\left(G^{-1}\left[\Sigma_0^{-1}\theta_0 + P^\mathsf{T}\Omega^{-1}q\right], \Sigma + G^{-1}\right).$$

**Remark 3.3** (Ground-Truth Limit). Corollary 3.1.1 accurately and precisely predict ground-truth asset returns with:

1. *Perfect information*: $\Omega \to 0$, equivalently $\Omega^{-1} \to \infty$.

2. *Accurate views*: $q \to Pr^\star$ where $r^\star$ is true asset returns.

3. *The $q \leftrightarrow F \leftrightarrow \theta$ linear model (Definition 3.2) reduces to the classical noisy views model (Black & Litterman, 1992)*: $(\alpha, \beta, \gamma) = (0, 0, 1)$, so the residual $(q - P\alpha - PF\beta) \to q$.

The M-BL posterior mean satisfies:

$$\lim_{\substack{\Omega \to 0 \\ q \to r^\star}} \mu_{\theta|q,\Omega,F,\Omega^F} = (G^M)^{-1} \Bigg[ \underbrace{P^\mathsf{T}\Omega^{-1}(Pr^\star - P\alpha - PF\beta)}_{\text{dominant term}}$$

$$+ \underbrace{\Sigma_0^{-1}\theta_0 + (\Omega^F)^{-1}(\alpha^F + F\beta^F)}_{\text{bounded}} \Bigg]$$

$$= (G^M)^{-1}[P^\mathsf{T}\Omega^{-1}Pr^\star + o(\Omega^{-1})]$$

$$= r^\star \quad \text{(a.s.)}$$

where the last step follows $G^M = \Sigma_0^{-1} + (\Omega^F)^{-1} + P^\mathsf{T}\Omega^{-1}P \to P^\mathsf{T}\Omega^{-1}P$, $(G^M)^{-1}o(\Omega^{-1}) \to 0$. As a result, Corollary 3.1.1 becomes

$$\widetilde{r} \xrightarrow{d} \delta_{r^\star} \quad \text{as} \quad \Omega \to 0, \ q \to Pr^\star$$

where $\delta_{r^\star}$ denotes the Dirac measure at $r^\star$.

Corollary 3.1.1 estimates asset returns under the general scenario where views are observed, i.e., solves Problem 2. It generalizes the classical Black-Litterman model, recovering its form when features are uninformative, and approaches the true returns with accurate and precise views (Remarks 3.2 and 3.3). With the predictive estimation of asset returns $\widetilde{r}$, the mean-variance optimization framework (Definition 2.1) determines the portfolio weights $w_{\text{M-BL}}$.

### 3.4 Scenario: Features with Latent Views

In the previous section, we solve Problem 2 with features and observed views. However, an investor using the Black-Litterman model may not be an expert at quantifying the views. In this section, we discuss the scenario without the views (Problem 3) by specifying two effects of the features introduced in Section 3.2 and treating $q$ and $\Omega$ as latent variables. Incorporating the effects into the network, we showcase two graphical forms in Figure 2 and define the Shared-Latent-Parametrization Black-Litterman (SLP-BL) and Feature-Influenced-Views Black-Litterman (FIV-BL) models (Definitions 3.4 and 3.6) based on them. Then, we estimate posterior distribution over both asset returns $r$ and their parameter $\theta$ (Corollary 3.1.1 and Theorem 3.1).

Consider the following problem:

**Problem 3** (Feature-Integrated Predictive Estimation). Let $r \in \mathbb{R}^m$ represent the returns of the $m$ assets and $\widetilde{r}$ denote the unobserved (or future) asset returns. Let $f_i \in \mathbb{R}^d$ represent the features of the $i$-th asset and $F \in \mathbb{R}^{m \times dm}$ be a block-diagonal matrix defined as

$$F := \text{diag}(f_1^\mathsf{T}, f_2^\mathsf{T}, \ldots, f_m^\mathsf{T}).$$

Given $D := (F, \Omega^F)$ where $F$ is estimated from observations of asset returns and features $\{(r_l, F_l)\}_{l=1}^n$ and $\Omega^F$ is the homoscedastic error matrix corresponding to the observations $\{F_l\}_{l=1}^n$, the goal is to estimate unobserved asset returns $\widetilde{r} \sim p(r|D)$.

We aim to solve Problem 3 by feature-integrated Black-Litterman network. We approach this by considering the two causal effects in Section 3.2 and treating the views and uncertainty matrix $(q, \Omega)$ as latent parameters. Specifically,

- **Effect 1**: Features $F$ are extracted from the parameter $\theta$. Consequently, the features $F$, along with their error term $\epsilon^F \sim N(0, \Omega^F)$, share the common parameter $\theta$ with asset returns $r$ and investor views $q$.

- **Effect 2**: Features $F$ influence the formation of views $q$. In this scenario, the features $F$, along with their error term $\epsilon^F \sim N(0, \Omega^F)$, are related to the latent views $q$ through a separate equation from the parameter $\theta$.

However, in this section, we do not mix the two effects.

**Remark 3.4** (Rationale of Differentiating Effect 1 and 2)**.** We differentiate the two effects because, in the scenario without investor views, the previous M-BL model (Definition 3.3) estimates the parameter $\theta$ directly by $(F, \Omega)$, meaning Effect 1 dominates over Effect 2 when both are present. If, in the general scenario where views are observed, Effect 2 is more significant than Effect 1, then, when views are latent, the ignorance of Effect 2 leads to biased estimation. This matches the intuition: if we select the features not directly related to the asset (Effect 1) but highly influence the investor views (Effect 2), such as macroeconomic indicators like interest rates or CPI, then using these features to estimate asset returns directly is biased. To avoid this bias, we differentiate the two effects with two modeling strategies. One handles the case where Effect 1 dominates, and the other handles the case where Effect 2 is more significant.

We showcase the feature-integrated Black-Litterman network as two configurations: one incorporating Effect 1 and another incorporating Effect 2. Intuitively, the first better captures generic features while the second more effectively handles the non-asset-related features.

This implies that, in practice, if an investor takes generic features of assets (e.g. indicators derived from the time series of each asset, as shown in our experiment), configuration 1 should be used. If an investor takes features not specific to individual assets (e.g. interest rates), configuration 2 should be used. The two configurations are not contradicting, so one can take both types of features and incorporate them correspondingly.

We visualize two configurations of the network in Figure 3 and define one model for each configuration accordingly.

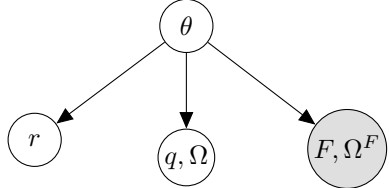

Configuration 1: Shared Latent Parametrization.

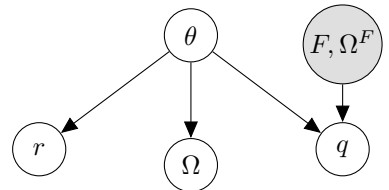

Configuration 2: Feature-Influenced Views.

Figure 3: Feature-Integrated Black-Litterman network with features and latent Views $(\theta, r, q, \Omega, F, \Omega^F)$.

CONFIGURATION 1: SHARED LATENT PARAMETRIZATION

To capture Effect 1, we follow the $\theta \leftrightarrow F$ relationship (Definition 3.1). By incorporating Effect 1, we showcase the feature-integrated Black-Litterman network as Shared-Latent-Parametrization Black-Litterman (SLP-BL) model:

**Definition 3.4** (SLP-BL model $(\theta, r, q, \Omega, F, \Omega^F)$)**.** Let $r \in \mathbb{R}^m$ be the returns of the $m$ assets, parametrized by $\theta$, with $r \sim p(r|\theta)$. Let $q \in \mathbb{R}^k$ represent the views on the returns of the $k$ specified portfolio and $\Omega \in \mathbb{R}^{k \times k}$ be the uncertainty matrix. Let $F \in \mathbb{R}^{m \times dm}$ be the features of the $m$ assets and error matrix $\Omega^F \in \mathbb{R}^{m \times m}$ The SLP-BL model is a portfolio model composed of four fundamental density functions:

1. Parametrized Asset Returns: $p(r|\theta)$, the distribution of asset returns given the parameter.

2. Prior: $\pi(\theta)$, representing market equilibrium.

3. Likelihood of Views: $L(\theta|q, \Omega) := p(q, \Omega|\theta)$, the relationship between the parameter and the views.

4. Likelihood of Features: $L(\theta|F, \Omega^F) := p(F, \Omega^F|\theta)$, the $\theta \leftrightarrow F$ relationship (Definition 3.1).

We show the posterior estimation on $\theta$ of the SLP-BL model:

**Theorem 3.2** (Parameter Estimation of the SLP-BL Model)**.** Given a SLP-BL model $(\theta, r, q, \Omega, F, \Omega^F)$ (Definition 3.4) and regression parameters $(\alpha^F, \beta^F) \in \mathbb{R}^m \times \mathbb{R}^{dm}$, assume

$$\theta \sim N(\theta_0, \Sigma_0), \tag{3.5}$$

$$\theta = \alpha^F + F\beta^F + \epsilon^F, \quad \epsilon^F \sim N(0, \Omega^F). \tag{3.6}$$

Define $G^F := \Sigma_0^{-1} + (\Omega^F)^{-1}$. The posterior mean is

$$p(\theta|F, \Omega^F) = N\left(\theta; (G^F)^{-1}\left[\Sigma_0^{-1}\theta_0 \\ + (\Omega^F)^{-1}(\alpha^F + F\beta^F)\right], (G^F)^{-1}\right).$$

*Proof.* See Appendix E.3 for a detailed proof. □

The posterior estimation of parameter $\theta$ enables a predictive estimation on $\widetilde{r}$:

**Corollary 3.2.1** (Predictive Estimation by the SLP-BL Model). Define $G^F := \Sigma_0^{-1} + (\Omega^F)^{-1}$. Assume $r \sim N(\theta, \Sigma)$. Then, under Theorem 3.2, SLP-BL model gives the predictive estimation of unobserved asset returns $\widetilde{r} := r|F, \Omega^F$ as

$$\widetilde{r} \sim N \left( G^F \left[ \Sigma_0^{-1}\theta_0 \right. \right.$$
$$\left. \left. +(\Omega^F)^{-1}(\alpha^F + F\beta^F) \right], \Sigma + (G^F)^{-1} \right).$$

**Remark 3.5** (Features Replace Views). Corollary 3.1.1 is equialvent to classical Black-Litterman model if:

1. *Features recover investor views*: $\alpha^F + F\beta^F \to P^{-1}q$

2. *Error matrix recovers uncertainty*: $(\Omega^F)^{-1} \to P^\mathsf{T}\Omega^{-1}P$

Under these conditions, we have $G^F \to \Sigma_0^{-1} + P^\mathsf{T}\Omega^{-1}P = G$, and thus recover (2.5):

$$r|q, \Omega \sim N \left( G^{-1} \left[ \Sigma_0^{-1}\theta_0 + P^\mathsf{T}\Omega^{-1}q \right], \Sigma + G^{-1} \right).$$

Corollary 3.2.1 estimates asset returns without the views, i.e., solves Problem 3. With the predictive estimation of asset returns $\widetilde{r}$, the mean-variance optimization framework (Definition 2.1) outputs the portfolio weights $w_{\text{SLP-BL}}$. See Appendix A for the selection of $\{\Sigma, \Sigma_0, \theta_0, \alpha^F, \beta^F, \Omega^F\}$.

CONFIGURATION 2: FEATURE-INFLUENCED VIEWS

To capture Effect 2, we define a $q \leftrightarrow F$ relationship by a multivariate linear model with local dependency:

**Definition 3.5** ($q \leftrightarrow F$ Linear Model). Given features $F \in \mathbb{R}^{m \times dm}$ and portfolio weight matrix for $k$ specified portfolios $P \in \mathbb{R}^{k \times m}$, let $q \in \mathbb{R}^k$ be the views on returns of the $k$ portfolios. Define regression intercept vector $\alpha \in \mathbb{R}^m$, regression coefficient vector $\beta \in \mathbb{R}^{dm}$, random error $\epsilon^F \in \mathbb{R}^m$, and error matrix $\Omega^F \in \mathbb{R}^{m \times m}$ such that:

$$q = P(\alpha + F\beta + \epsilon^F), \quad \epsilon^F \sim N(0, \Omega^F),$$

Furthermore, define $\beta_1, \beta_2, \ldots, \beta_m \in \mathbb{R}^d$ as $m$ partitions of the vector $\beta$ such that:

$$\beta = \left[ \beta_1^\mathsf{T}, \beta_2^\mathsf{T}, \ldots, \beta_m^\mathsf{T} \right]^\mathsf{T}.$$

This captures the relationship without the loss of generality:

**Remark 3.6** (Noisy Implied Asset Returns). Based on the intuition that the views $q$ are formed on the returns of the $k$ specified portfolios, define the noisy implied asset returns

$$r^F := \alpha + F\beta + \epsilon^F$$

such that: $q = Pr^F$. Then, the dependency between each element of this implied asset returns and $d$ features becomes local:

$$r_i^F = \alpha_i + F_{i,:}\beta + \epsilon_i^F = \alpha_i + \beta_i^\mathsf{T} f_i + \epsilon_i^F, \quad i \in [m].$$

Remark 3.6 allows estimations of regression parameters and the error matrix $(\alpha, \beta, \Omega^F)$ based on the observations $\{(r_l, F_l)\}_{l=1}^n$. See Appendix A for details.

We now introduce the final piece in this configuration: Regarding the uncertainty matrix $\Omega$, a simplified assumption is that when an investor forms views based on features, the error matrix $\Omega^F$ captures all the information about this uncertainty. This would suggest omitting $\Omega$ from our model due to the replacement with $\Omega^F$. Yet, in the general case, $\Omega$ remains necessary as it represents the intrinsic uncertainty of the views, regardless of the $(F, \Omega^F)$[2]. Additionally, retaining $\Omega$ allows our model to remain applicable when both $(q, \Omega)$ and $F$ are observed. To treat $\Omega$ as a latent parameter, a prior $\pi(\Omega)$ must be specified to enable Bayesian inference.

By incorporating Effect 2 of the features characterized by the $q \leftrightarrow F$ linear models (Definition 3.5) and a given prior $\pi(\Omega)$, we showcase the feature-integrated Black-Litterman network as the following Feature-Influenced-Views Black-Litterman (FIV-BL) model:

**Definition 3.6** (FIV-BL model $(\theta, r, q, \Omega, F, \Omega^F)$). Let $r \in \mathbb{R}^m$ be the returns of the $m$ assets, parametrized by $\theta$, with $r \sim p(r|\theta)$. Let $q \in \mathbb{R}^k$ represent the views on the returns of the $k$ specified portfolio and $\Omega \in \mathbb{R}^{k \times k}$ be the uncertainty matrix. Let $F \in \mathbb{R}^{m \times dm}$ be the features of the $m$ assets and error matrix $\Omega^F \in \mathbb{R}^{m \times m}$ The FIV-BL model is a portfolio model composed of five fundamental density functions:

1. Parametrized Asset Returns: $p(r|\theta)$, the distribution of asset returns given the parameter.

2. Prior: $\pi(\theta)$, representing market equilibrium.

3. Likelihood of Views: $L(\theta|q, \Omega) := p(q, \Omega|\theta)$, the relationship between the parameter and the views.

4. Views given Features: $p(q|F, \Omega^F)$, the $q \leftrightarrow F$ relationship (Definition 3.5).

5. Prior on Uncertainty Matrix: $\pi(\Omega)$, representing intrinsic uncertainty of the views.

The FIV-BL model marginalizing out latent parameters $(q, \Omega)$ to estimate the posterior of $\theta$:

**Theorem 3.3** (Parameter Estimation of the FIV-BL Model). Let $P \in \mathbb{R}^{k \times m}$ be the portfolio weight matrix for $k$ specified portfolios. Given a FIV-BL model $(\theta, r, q, \Omega, F, \Omega^F)$ (Definition 3.6), regression parameters $(\alpha, \beta) \in \mathbb{R}^m \times \mathbb{R}^{dm}$, and a prior $\pi(\Omega)$, assume

$$\theta \sim N(\theta_0, \Sigma_0) \tag{3.7}$$

$$P\theta = q + \epsilon, \quad \epsilon \sim N(0, \Omega), \tag{3.8}$$

$$q = P(\alpha + F\beta + \epsilon^F), \quad \epsilon^F \sim N(0, \Omega^F), \tag{3.9}$$

---

[2]This concept is similar to the existence of the intrinsic covariance $\Sigma$ regardless of the prior parameter $(\theta_0, \Sigma_0)$ of $\theta$ in Lemmas 2.1 and C.1.

where $\theta_0 \in \mathbb{R}^m$ and $\Sigma_0 \in \mathbb{R}^{m \times m}$ are given prior mean and covariance, and $(\epsilon, \epsilon^F)$ are mutually independent. Define $G := \Sigma_0^{-1} + P^\mathsf{T} \Omega^{-1} P$. The posterior mean distribution is:

$$p(\theta | F, \Omega^F) = \int N\left(\theta; \mu_{\theta|\Omega,F,\Omega^F}, \Sigma_{\theta|\Omega,F,\Omega^F}\right) \pi(\Omega) d\Omega,$$

(3.10)

where

$$\begin{cases} \mu_{\theta|\Omega,F,\Omega^F} = G^{-1}\left(\Sigma_0^{-1}\theta_0 + P^\mathsf{T}\Omega^{-1}P(\alpha + F\beta)\right), \\ \Sigma_{\theta|\Omega,F,\Omega^F} = G^{-1} + G^{-1}P^\mathsf{T}\Omega^{-1}\left(P\Omega^F P^\mathsf{T}\right)\Omega^{-1}PG^{-1}. \end{cases}$$

*Proof.* See Appendix E.4 for a detailed proof. □

**Remark 3.7** (Posterior Collapse Under Perfect Views)**.** If we omit the intrinsic uncertainty matrix by $\Omega \to 0$, or equivalently $\Omega^{-1} \to 0$, we have

$$G \to P^\mathsf{T}\Omega^{-1}P,$$

$$\mu_{\theta|\Omega,F,\Omega^F} \to G^{-1}\left(P^\mathsf{T}\Omega^{-1}P(\alpha + F\beta)\right),$$

$$\Sigma_{\theta|\Omega,F,\Omega^F} \to G^{-1} + G^{-1}P^\mathsf{T}\Omega^{-1}\left(P\Omega^F P^\mathsf{T}\right)\Omega^{-1}PG^{-1}.$$

Thus, the posterior collapses to

$$\theta|F, \Omega^F = \theta|\Omega, F, \Omega^F \sim N(\alpha + F\beta, \Omega^F),$$

effectively recovering the $\theta \leftrightarrow F$ relationship $\theta = \alpha + F\beta + \epsilon^F$ (Definition 3.1) except losing the prior information on $\theta$. Furthermore, reintroducing this prior leads to Theorem 3.2.

The integral (3.10) is a form of Infinite Gaussian Mixture model (IGMM) (Rasmussen, 1999). In general, there is no further closed-form solution for it unless $\Omega$ is restricted to a special conjugate family or effectively collapses to a point mass (i.e., $\Omega$ is known and fixed). In non-conjugate settings, the expression remains a continuous mixture of Gaussian distributions, and must be evaluated or approximated numerically (e.g. via Monte Carlo or approximation methods (Newman & Barkema, 1999; Kruschke, 2010; Wainwright et al., 2008; Blei et al., 2017)).

Since there is no trivial conjugate prior $\pi(\Omega)$ for the likelihood $\theta|\Omega, F, \Omega^F$, here we offer an approximation method. We first substitute $\Omega$ with $\Sigma_{\theta|\Omega,F,\Omega^F}$. Then, we approximate the mean of the likelihood $\mu_{\theta|\Omega,F,\Omega^F}$ as a constant. Finally, we assign a conjugate prior to $\Sigma_{\theta|\Omega,F,\Omega^F}$ as an Inverse-Wishart (IW) distribution. This allows us to obtain a tractable joint distribution $p(\theta, F, \Omega^F)$ — specifically a Normal-Inverse-Wishart (NIW) distribution. As a result, the posterior mean distribution $p(\theta|F, \Omega^F)$ follows a student-t distribution after marginalizing out $\Sigma_{\theta|\Omega,F,\Omega^F}$:

**Corollary 3.3.1** (Conjugate Prior)**.** Consider a FIV-BL model $(\theta, r, q, \Omega, F, \Omega^F)$ (Definition 3.6) with constants $(P, \Sigma, \theta_0, \Sigma_0, \alpha, \beta, \Omega_0) \in \mathbb{R}^{k \times m} \times \mathbb{R}^{m \times m} \times \mathbb{R}^m \times \mathbb{R}^{m \times m} \times \mathbb{R}^m \times \mathbb{R}^{dm} \times \mathbb{R}^{k \times k}$. Define $G := \Sigma_0^{-1} + P^\mathsf{T}\Omega^{-1}P$. Assume

$$\theta | \Omega, F, \Omega^F \sim N(\mu', \Sigma'),$$

where $\begin{cases} \mu' := G^{-1}\left(\Sigma_0^{-1}\theta_0 + P^\mathsf{T}\Omega_0^{-1}P[\alpha + F\beta]\right), \\ \Sigma' := G^{-1} + G^{-1}P^\mathsf{T}\Omega^{-1}\left(P\Omega^F P^\mathsf{T}\right)\Omega^{-1}PG^{-1}. \end{cases}$

Assume $\Sigma'$ have an Inverse-Wishart prior:

$$\pi(\Sigma') = IW(\Sigma'; \Psi', \nu'), \quad (\Psi', \nu') \in \mathbb{R}^m \times \mathbb{R} \quad (3.11)$$

Then the marginal posterior of $\theta$ given $(F, \Omega^F)$ follows a multivariate-$t$ distribution:

$$p(\theta | F, \Omega^F) \sim t_{\nu'}\left(\theta; \mu', \frac{\Psi'}{\nu' - m + 1}\right).$$

*Proof.* See Appendix E.5 for a detailed proof. □

Since $t$-distribution lacks a conjugate prior, we omit intrinsic covariance $\Sigma$ in estimating unobserved asset returns $\widetilde{r}$:

**Corollary 3.3.2** (Approximated Predictive Estimation by the FIV-BL Model)**.** Assume $r \sim N(\theta, \Sigma)$. Under Theorem 3.3, considering $\theta|\Omega, F, \Omega^F \sim N(\mu', \Sigma')$, assume:

$$\Sigma' \sim IW(\Psi', \nu'),$$

$\mu'$ is a constant $G^{-1}\left(\Sigma_0^{-1}\theta_0 + P^\mathsf{T}\Omega_0^{-1}P(\alpha + F\beta)\right)$.

Then, as $\Sigma \to 0$, FIV-BL model gives the predictive estimation $\widetilde{r} := r|F, \Omega^F$ as

$$\widetilde{r} \sim t_{\nu'}\left(\widetilde{r}; \mu', \frac{\Psi'}{\nu' - m + 1}\right).$$

Corollary 3.3.2 also estimates asset returns without the views, i.e., solves Problem 3. With the predictive estimation of asset returns $\widetilde{r}$, the mean-variance optimization framework (Definition 2.1) determines the portfolio weights $w_{\text{FIV-BL}}$. See Appendix A for the selection of $\{\Sigma, \Sigma_0, \theta_0, P, \alpha, \beta, \Omega^F, \Psi', \nu', \Omega_0\}$.

## 4  Proof-of-Concept Experiments

Depart from the classic Black-Litterman model that relies on subjective investor views, our model estimates the posterior distribution over asset returns directly from the feature data. To demonstrate this concept, we consider the setting without subjective investor views. Specifically, we focus on integrating asset-specific features, as discussed in Remark 3.4, and choose the SLP-BL model (Definition 3.4) accordingly. We show the model works under this setting and consistently outperforms the benchmarks.

**Dataset I: SPDR Sector ETFs.** We collect adjusted daily closing prices and volume for 11 Sector ETFs (Table 4) from April 13, 2004 to February 22, 2024 (20 years). To avoid selection bias, the portfolio selection list is updated in sync with the introduction of new sectors.

**Dataset II: Dow Jones Index.** We collect adjusted daily closing prices and volume for 41 stocks (Table 5) that have been part of the Dow Jones index from January 5, 1994 to

February 22, 2024 (30 years). To avoid selection bias, the portfolio selection list is updated in sync with the index.

**Backtest Task.** We backtest our SLP-BL model for each dataset period. On each monthly rebalance day, the model outputs a portfolio weight $w_{\text{SLP}-\text{BL}}$ that maximizes the Sharpe ratio (Definition F.1), a standardized mean-variance optimization framework (Definition 2.1). In the model, the prior is set as traditional Markowitz model and the features are selected based on nine generic indicators (Table 3) derived from asset-specific data. We follow Appendix A for the choice of $\{\Sigma, \Sigma_0, \theta_0, \alpha^F, \beta^F, \Omega^F\}$ except that, to avoid the issues of mismatch scale, we use price and indicators data to derive the regression parameters.

**Benchmarks and Evaluation.** The benchmarks of our portfolio model are set as (i) market index (e.g. S&P500 and DJIA) (ii) equal-weighted portfolio model (iii) traditional Markowitz model. We evaluate the models with the following metrics: Cumulative Return, Compound Annual Growth Rate (CAGR), Sharpe Ratio, Maximum Drawdown, and Volatility. We present the results for five pairs of traditional Markowitz model and our Black-Litterman model with varying rolling window lengths of historical returns: 50 days, 80 days, 100 days, 120 days, and 150 days.

**Results.** The SLP-BL model consistently outperforms both traditional Markowitz model and market indices across two datasets (Tables 1, 2 and 6 and Figures 4, 5, 8 and 9). This is attributed to the more stable portfolio weights based on Bayesian framework, as shown by Figures 6 and 7.

Table 1: Performance on SPDR Sector ETFs Dataset.

| | Cumulative Return (%) ↑ | CAGR (%) ↑ | Sharpe Ratio ↑ | Max Drawdown (-%) ↓ | Volatility (%/ann.) ↓ |
|---|---|---|---|---|---|
| EQW | 450.74 | 6.11 | 0.61 | 44.90 | 16.40 |
| S&P500 | 545.77 | 6.69 | 0.59 | 55.19 | 19.03 |
| MV (50d) | 134.12 | 3.00 | 0.35 | 53.11 | 15.99 |
| BL (50d) | 541.99 | 6.67 | 0.66 | 46.56 | 16.24 |
| MV (80d) | 291.21 | 4.85 | 0.50 | 38.58 | 16.33 |
| BL (80d) | 609.66 | 7.04 | 0.69 | 46.78 | 16.12 |
| MV (100d) | 411.83 | 5.84 | 0.57 | 36.37 | 16.91 |
| BL (100d) | 602.75 | 7.01 | 0.70 | 46.05 | 15.91 |
| MV (120d) | 412.87 | 5.84 | 0.57 | 36.10 | 17.12 |
| BL (120d) | 587.50 | 6.93 | 0.70 | 46.11 | 15.74 |
| MV (150d) | 249.11 | 4.44 | 0.45 | 47.49 | 17.37 |
| BL (150d) | 556.13 | 6.75 | 0.68 | 44.54 | 15.91 |

## 5   Discussion and Conclusion

We propose a Bayesian reformulation of the Black-Litterman model for portfolio optimization without the need for subjective investor views. Our key contribution is a unified Bayesian network that integrates features and infers parameters. In the case of observed views (Problem 2), the network estimates asset returns based on a mix of two feature effects (Theorem 3.1, Corollary 3.1.1), generalizing the classical Black-Litterman model and recovering ground-truth estimation with perfect views (Remark 3.3). In the case

Table 2: Performance on Dow Jones Index Dataset.

| | Cumulative Return (%) ↑ | CAGR (%) ↑ | Sharpe Ratio ↑ | Max Drawdown (-%) ↓ | Volatility (%/ann.) ↓ |
|---|---|---|---|---|---|
| EQW | 4,606.66 | 9.22 | 0.75 | 58.90 | 19.54 |
| DJIA | 932.51 | 5.49 | 0.52 | 53.78 | 17.97 |
| MV (50d) | 774.08 | 5.09 | 0.45 | 63.35 | 20.83 |
| BL (50d) | 3,980.23 | 8.86 | 0.78 | 42.42 | 17.84 |
| MV (80d) | 1,081.47 | 5.82 | 0.51 | 53.98 | 20.26 |
| BL (80d) | 4,603.82 | 9.22 | 0.84 | 39.95 | 16.81 |
| MV (100d) | 1,529.60 | 6.60 | 0.55 | 56.06 | 20.59 |
| BL (100d) | 4,557.03 | 9.19 | 0.85 | 39.92 | 16.56 |
| MV (120d) | 1,577.61 | 6.67 | 0.57 | 46.73 | 20.07 |
| BL (120d) | 4,819.83 | 9.33 | 0.87 | 39.81 | 16.42 |
| MV (150d) | 2,208.84 | 7.45 | 0.62 | 41.02 | 20.03 |
| BL (150d) | 3,405.78 | 8.49 | 0.80 | 40.42 | 16.51 |

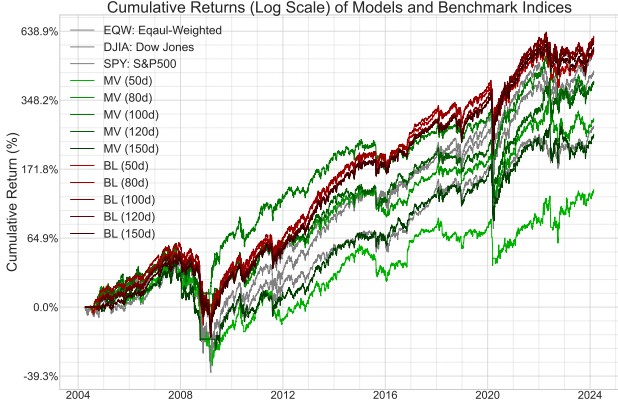

Figure 4: Cumulative Return on SPDR Sectors ETFs Dataset.

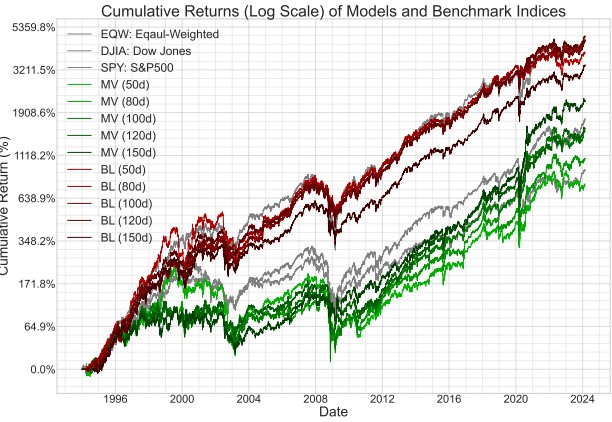

Figure 5: Cumulative Return on Dow Jones Index Dataset.

of latent views (Problem 3), we differentiate the feature effects to handle distinct features (Remark 3.4). Accordingly, we present two models: the first provides closed-form asset return estimation (Theorem 3.2, Corollary 3.2.1), while the second results in a mixture model that requires numerical methods (Theorem 3.3, Corollary 3.3.1, Corollary 3.3.2). Numerically, our model works without investor views and demonstrates consistent, hyperparameter-robust improvements over the Markowitz model and market indices across long-term, real-world datasets (Section 4).

## Impact Statement

This work improves portfolio optimization by reducing subjective human inputs. It enhances transparency and promotes data-driven decision-making. The framework benefits both institutional and individual investors with more reliable and fair strategies. However, data-driven models may amplify biases, so careful evaluation is needed for fair outcomes. Overall, this work advances financial modeling and emphasizes ethical implementation.

## Acknowledgments

TL would like to thank Gamma Paradigm Research and NTU ABC Lab for support. JH would like to thank Han Liu, Mimi Gallagher, Sara Sanchez, Dino Feng and Andrew Chen for enlightening discussions on related topics, and the Red Maple Family for support. The authors would like to thank the anonymous reviewers and program chairs for constructive comments. JH is supported by the Northwestern University. The content is solely the responsibility of the authors and does not necessarily represent the official views of the funding agencies.

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

# Appendix

# A   Hyperparameter Selections

Here we provide a practical guide to derive hyperparameter set $\{\Sigma, \Sigma_0, \theta_0, \alpha^F, \beta^F, \Omega^F\}$ for Shared-Latent-Parametrization Black-Litterman model (SLP-BL model, Definition 3.4) and $\{\Sigma, \Sigma_0, \theta_0, P, \alpha, \beta, \Omega^F, \Psi', \nu', \Omega_0\}$ for Feature-Influenced-Views Black-Litterman model (FIV-BL model, Definition 3.6)

Among the entries of $\{\Sigma, \Sigma_0, \theta_0\}$, a practitioner first approximate $\Sigma$ using sample variance given $\{r_l\}_{l=1}^n$ and let $\Sigma_0 = \tau\Sigma$ be a matrix proportional to the covariance matrix, following (Salomons, 2007). Numerous papers (Black & Litterman, 1992; Lee, 2000; Ellison, 2004; Idzorek, 2007; Salomons, 2007) address the choice of the scaling factor $\tau$, mostly suggesting a constant in $(0, 1]$. Given the approximated $\Sigma$ and $\Sigma_0$, one obtains $\theta_0$ by Lemma C.1.

In this work, we take $P = I$ as a $m \times m$ identity matrix because the features $F$ are asset-specific (Problems 2 and 3). Given the historical observations $\{(r_l, F_l)\}_{l=1}^n$ in (Problems 2 and 3), define:

$$\bar{r} := \frac{1}{n}\sum_{l=1}^n r_l, \quad \widetilde{r}_l := r_l - \bar{r}, \quad \overline{F} := \frac{1}{n}\sum_{l=1}^n F_l, \quad \widetilde{F}_l := F_l - \overline{F}.$$

The following context suggests how $\{(r_l, F_l)\}_{l=1}^n$ enables estimating $\{\widehat{\Omega}^F, \widehat{\alpha}^F, \widehat{\beta}^F, \widehat{\alpha}, \widehat{\beta}\}$. We estimate the error matrix $\widehat{\Omega}^F$ based on kernel density estimation on every feature. Specifically, consider a rule-of-thumb bandwidth parameter (Silverman, 2018):

$$h = \left(\frac{4}{dm + 2}\right)^{\frac{2}{dm+4}} n^{-\frac{2}{dm+4}}$$

where $m$ is the number of assets, $d$ the number of features for each asset, and $n$ the sample size.

Recall that $f_i \in \mathbb{R}^d$ represent the features on the $i$-th asset and be part of the features $F \in \mathbb{R}^{m \times dm}$:

$$F := \mathrm{diag}(f_1^\mathsf{T}, f_2^\mathsf{T}, \ldots, f_m^\mathsf{T}).$$

we scale element-wise variances of $f_i$, or $\mathrm{Var}(f_{i,j})$ for $j$-th feature of the $i$-th asset by $h$ to construct the diagonal matrix

$$\widetilde{H} = \mathrm{diag}(h \cdot \mathrm{Var}(f_{1,1}), \ldots, h \cdot \mathrm{Var}(f_{m,d})).$$

Then, for each asset return $r_i$, we compute the ordinary least squares (OLS) coefficients $B_i$ and intercept $a_i$ by predictors $f_i$[3]. These coefficients are aggregated into a block-diagonal matrix $B \in \mathbb{R}^{m \times dm}$. The covariance estimate is $\widehat{\Omega}^F = B\widetilde{H}B^\mathsf{T}$.

For $\{\widehat{\alpha}^F, \widehat{\beta}^F\}$, we conduct maximum likelihood estimation based on the $\theta - F$ model (Definition 3.1) and $r \sim N(\theta, \Sigma)$. By Lemma D.4, we obtain

$$\widehat{\alpha}^F = \bar{r} - \overline{F}\widehat{\beta}^F, \quad \widehat{\beta}^F = \left(\sum_{l=1}^n \widetilde{F}_l^\top (\widehat{\Omega}^F + \Sigma)^{-1}\widetilde{F}_l\right)^{-1}\sum_{l=1}^n \widetilde{F}_l^\top (\widehat{\Omega}^F + \Sigma)^{-1}\widetilde{r}_l.$$

For $\{\widehat{\alpha}, \widehat{\beta}\}$, we conduct maximum likelihood estimation in the $q - F$ model (Definition 3.5), assuming that the observed returns $r_l$ are samples from the noisy implied asset returns $r^F$ defined in Remark 3.6. By Lemma D.4, we obtain

$$\widehat{\alpha} = \bar{r} - \overline{F}\widehat{\beta}, \quad \widehat{\beta} = \left(\sum_{l=1}^n \widetilde{F}_l^\top (\widehat{\Omega}^F)^{-1}\widetilde{F}_l\right)^{-1}\sum_{l=1}^n \widetilde{F}_l^\top (\widehat{\Omega}^F)^{-1}\widetilde{r}_l.$$

For $\{\Psi', \nu', \Omega_0\}$, the conjugate prior parameters $(\Psi', \nu')$ in $\Sigma' \sim IW(\Psi', \nu')$ have distinct roles: $\Psi'$ encodes prior knowledge about the covariance shape and scale, acting as a "pseudo-covariance matrix" with mean $\mathbb{E}[\Sigma'] = \Psi'/(\nu' - m + 1)$ (if $\nu' > m - 1$). Larger $\Psi'$ implies stronger prior beliefs about higher covariances. The degrees of freedom $\nu'$ control prior strength, functioning as an effective sample size. Smaller $\nu'$ allows the data to dominate, while larger $\nu'$ enforces $\Psi'$. For a weakly informative prior, the rule of thumb is to set $\Psi' = I$ (minimal informative scale matrix) and $\nu' = m + 2$, assuming no strong prior correlations. Gelman et al. (1995); Hoff (2009); Murphy (2012) discuss details on this topic.

Selecting $(\Psi', \nu')$ allows us to compute the approximated constant $\Omega_0$. Since $\Sigma'$ is a deterministic function of $(\Omega, F, \Omega^F)$, $\Omega$ is likewise a deterministic function of $(\Sigma', F, \Omega^F)$. Therefore, we let $\Omega_0 = \Omega(\Sigma'_0, F, \Omega^F)$ where $\Sigma'_0 = \Psi'/(\nu' - m + 1)$ is the mean of the distribution $IW(\Psi', \nu')$.

---

[3]One may adjust the targetted data to avoid the issues of misaligned scale. For example, price versus momentum indicators.

# B  Related Work

## B.1  Bayesian Portfolio Optimization

**Why Bayesian?**  To address the parameter estimation risk in traditional portfolio optimization shown by (Markowitz, 1952; Kalymon, 1971), Barry (1974); Klein & Bawa (1976); Brown (1976) advocate Bayesian framework upon prior information in portfolio optimization. Foundational works by (Jorion, 1986) and (Black & Litterman, 1992) demonstrate how Bayesian shrinkage improves covariance estimation, reducing overfitting and highly sensitive weight in Markowitz-style allocations (Meucci, 2005; DeMiguel et al., 2009). Subsequent studies on robust Bayesian portfolio optimization include multiple approaches such as uncertainty estimation (Qiu et al., 2015; Yang et al., 2015), alternative prior specifications (e.g., heavy-tailed or non-conjugate priors) (Garlappi et al., 2007; Tu & Zhou, 2010; 2011), advanced sampling methods (Michaud & Michaud, 2007; Huang et al., 2021), and regularized optimization considering transaction costs (Olivares-Nadal & DeMiguel, 2018).

**Issues Around the Bayesian Framework.**  While these methods leverage analytical tractability to incorporate historical data or expert views (Ulf & Raimond, 2006), they often rely on restrictive assumptions (e.g., conjugate priors) or subjective expert inputs. Recent advancements, such as Markov chain Monte Carlo (MCMC) methods (Greyserman et al., 2006), relax these constraints, enabling inference in more complex hierarchical or time-series models. Meanwhile, contemporary approaches increasingly emphasize data-driven techniques for deriving expert inputs, including investor views in the Black-Litterman model (Black & Litterman, 1992).

## B.2  Data-Driven Black-Litterman Model

**Why Data-Driven?**  Across decades, the heuristic framework for deriving investor views $(q, \Omega)$ in the Black-Litterman model attracts research working on estimating investor views (Beach & Orlov, 2007; Palomba, 2008; Duqi et al., 2014; Silva et al., 2017; Deng, 2018; Kara et al., 2019; Kolm & Ritter, 2021; Teplova et al., 2023). Early efforts to estimate $(q, \Omega)$ employ historical return data within GARCH frameworks, framing view derivation as a time series prediction task (Beach & Orlov, 2007; Palomba, 2008; Duqi et al., 2014), but financial time-series data often exhibit high noise and insufficient signal-to-noise ratios for reliable prediction (Gómez & Maravall Herrero, 1998; Christensen & Li, 2014).

**Advancement in Generating Views.**  Recent advances mitigate the previously mentioned weaknesses in time series forecasting by integrating econometric models and machine learning — e.g., GARCH with neural networks (Bildirici & Ersin, 2009) or LSTM (Kim & Won, 2018), support vector machines (Pérez-Cruz et al., 2003; Kara et al., 2019), grey systems (Huang & Jane, 2009), and feature programming (Reneau et al., 2023). To embrace richer information, recent studies incorporate external data sources such as macroeconomic indicators (Zhou, 2009; Cheung, 2013), factors (Geyer & Lucivjanská, 2016; Kolm & Ritter, 2017; 2021).

**Neglected Issues from a Whole Perspective.** However, while these advanced methods achieve high forecast accuracy in isolation, errors can propagate through subsequent optimization pipelines when estimators $(q, \Omega)$ are naively embedded in the Black-Litterman framework. Finkel et al. (2006) addresses such issues of error propagation in multi-stage pipelines. Furthermore, in some works, estimating $q$ and $\Omega$ independently risks misaligned confidence assumptions. Without joint modeling, overconfidence in views (low $\Omega$) might amplify errors in $q$, and thus distort portfolio weights. Guo et al. (2017); Kendall & Gal (2017) discuss such confidence calibration and uncertainty estimation.

**Introduction of Bayesian Network.** Meanwhile, prior work has explored the use of Bayesian networks for the Black-Litterman model, which serve distinct purposes such as transferring the approach to factor models (Kolm & Ritter, 2017; 2021), addressing multiple expert views (Chen & Lim, 2020), or generalizing to multi-period frameworks (Abdelhakmi & Lim, 2024). Yet, these approaches still rely on human experts to specify the parameters $(q, \Omega)$.

**Our Work.**  To bridge these gaps—subjective inputs, error propagation, and incoherent estimation — we propose our Bayesian network reformulation of the Black-Litterman model. This framework unifies historical or external features and latent investor views $(q, \Omega)$ into a single Bayesian network, enabling inference over parameters (Theorems 3.2 and 3.3) and asset returns (Corollaries 3.2.1 and 3.3.2) directly from data. This eliminates reliance on heuristic inputs or disjointed estimators, ensuring coherent estimation and fully data-driven portfolio optimization.

# C    Supplementary Theoretical Backgrounds

## C.1    Prior and Likelihood of the Black-Litterman-Bayes model

Here we show how to model the prior and likelihood in the Black-Litterman-Bayes model (Definition 2.2). To obtain the prior $\pi(\theta)$, an investor sets a market portfolio and lets the prior be the market portfolio estimation. In the absence of views $q$, the BLB model reduces to this market portfolio, producing an estimation (exactly the prior) on asset returns and outputting a market portfolio weight $w_{\mathrm{market}}$[4] by Definition 2.1. For example, if the investor takes the traditional Markowitz model as the market portfolio, it would produce a normal distribution of the historical asset returns as the prior $\pi(\theta)$.

A commonly used market portfolio is market capitalization-weighted portfolio[5]. In this case, the market portfolio weight is the market capitalization weight $w_{\mathrm{cap}}$[6]:

**Assumption C.1** (Market Capitalization Equilibrium Prior)**.** In the absence of views $q$, the BLB model (Definition 2.2) produces an estimation of asset returns $\widetilde{r}$ such that the mean-variance optimization framework (Definition 2.1) has an optimal argument $w^{\star} = w_{\mathrm{cap}}$. In other words, $\widetilde{r}$ satisfied:

$$\operatorname*{argmax}_{w} \left\{ w^T \mathbb{E}[\widetilde{r}] - \frac{\delta}{2} w^T \operatorname{Var}[\widetilde{r}] w \right\} = w_{\mathrm{cap}}, \tag{C.1}$$

where $\delta \in [0, \infty]$ is a given risk-adjusted coefficient.

With Assumption C.1, we use a reverse optimization technique to derive the prior:

**Lemma C.1** (Reverse Optimization for Prior, page 139 of (Satchell & Scowcroft, 2007))**.** Let $r \in \mathbb{R}^m$ be $m$ asset returns, parametrized by $\theta$, with $r \sim p(r|\theta)$. Let the market capitalization weight on the $m$ assets be $w_{\mathrm{cap}} \in \mathbb{R}^m$ and $\delta \in [0, \infty]$ be a risk-adjusted coefficient. Assume

$$r \sim N(\theta, \Sigma), \quad \theta \sim N(\theta_0, \Sigma_0),$$

where $\Sigma, \Sigma_0 \in \mathbb{R}^{m \times m}$ are given intrinsic and prior covariance. Then the prior mean is

$$\theta_0 = \delta(\Sigma + \Sigma_0) w_{\mathrm{cap}}. \tag{C.2}$$

To obtain the likelihood function $L(\theta|q)$, we assume a probabilistic relationship between parameter $\theta$ and views $q$:

**Assumption C.2** (Classical Noisy Views Model, page 35 of (Black & Litterman, 1992))**.** Let $r \in \mathbb{R}^m$ be the returns of the $m$ assets, parametrized by $\theta$, with $r \sim p(r|\theta)$. Let $P \in \mathbb{R}^{k \times m}$ be the specified portfolio weight matrix. Let $q \in \mathbb{R}^k$ represent the views on the returns of the $k$ specified portfolio and $\Omega \in \mathbb{R}^{k \times k}$ be the uncertainty matrix. Assume

$$P\theta = q + \epsilon, \quad \epsilon \sim N(0, \Omega).$$

Under Assumptions C.1 and C.2, we can derive the Black-Litterman formula (Theorem 2.1) as the posterior estimation on $\theta$ of the Black-Litterman-Bayes model (Definition 2.2).

# D    Axillary Lemmas

## D.1    Integral of the Product of Two Gaussian Distributions

**Lemma D.1** (Integral of the Product of Two Gaussian Distributions, page 266 of (Aroian, 1947))**.** Let $x \in \mathbb{R}^n$, and let $N(x; \mu_1, \Sigma_1)$ and $N(x; \mu_2, \Sigma_2)$ be two multivariate Gaussian distributions with means $\mu_1, \mu_2 \in \mathbb{R}^n$ and positive definite covariance matrices $\Sigma_1, \Sigma_2 \in \mathbb{R}^{n \times n}$, respectively. Then, the integral of their product over $\mathbb{R}^n$ is given by:

$$\int N(x; \mu_1, \Sigma_1) N(x; \mu_2, \Sigma_2) dx = N(\mu_1; \mu_2, \Sigma_1 + \Sigma_2). \tag{D.1}$$

*Proof.* The product of two Gaussian PDFs is

$$N(x; \mu_1, \Sigma_1) N(x; \mu_2, \Sigma_2) = \frac{1}{(2\pi)^n |\Sigma_1|^{1/2} |\Sigma_2|^{1/2}} \exp\left(-\frac{1}{2} Q\right), \tag{D.2}$$

---

[4]Idzorek (2007) names it Implied Equilibrium Return Vector.

[5]A market capitalization-weighted portfolio performs a market capitalization-weighted index (e.g., S&P 500).

[6]The weight vector proportional to each asset's market cap.

where $Q = (x - \mu_1)^\top \Sigma_1^{-1}(x - \mu_1) + (x - \mu_2)^\top \Sigma_2^{-1}(x - \mu_2)$.

Expanding and combining terms:

$$Q = x^\top(\Sigma_1^{-1} + \Sigma_2^{-1})x - 2x^\top(\Sigma_1^{-1}\mu_1 + \Sigma_2^{-1}\mu_2) + c$$
$$= (x - A^{-1}b)^\top A(x - A^{-1}b) - b^\top A^{-1}b + c,$$

with $A = \Sigma_1^{-1} + \Sigma_2^{-1}$, $b = \Sigma_1^{-1}\mu_1 + \Sigma_2^{-1}\mu_2$, and $c = \mu_1^\top \Sigma_1^{-1}\mu_1 + \mu_2^\top \Sigma_2^{-1}\mu_2$.

Substituting back to (D.2):

$$N(x; \mu_1, \Sigma_1)N(x; \mu_2, \Sigma_2)$$
$$= \frac{1}{(2\pi)^n |\Sigma_1|^{1/2}|\Sigma_2|^{1/2}} \exp\left(-\frac{1}{2}(x - A^{-1}b)^\top A(x - A^{-1}b) + \frac{1}{2}b^\top A^{-1}b - \frac{1}{2}c\right).$$

Integrate over $x \in \mathbb{R}^n$:

$$\int_{\mathbb{R}^n} N(x; \mu_1, \Sigma_1)N(x; \mu_2, \Sigma_2)dx = \frac{(2\pi)^{n/2}|A^{-1}|^{1/2}}{(2\pi)^n |\Sigma_1|^{1/2}|\Sigma_2|^{1/2}} \exp\left(\frac{1}{2}b^\top A^{-1}b - \frac{1}{2}c\right). \tag{D.3}$$

Using $|A^{-1}| = \frac{|\Sigma_1||\Sigma_2|}{|\Sigma_1 + \Sigma_2|}$, we have

$$\frac{|A^{-1}|^{1/2}}{|\Sigma_1|^{1/2}|\Sigma_2|^{1/2}} = \frac{1}{|\Sigma_1 + \Sigma_2|^{1/2}}. \tag{D.4}$$

Simplify the exponential term in (D.3) using the identity:

$$\frac{1}{2}b^\top A^{-1}b - \frac{1}{2}c = -\frac{1}{2}(\mu_1 - \mu_2)^\top(\Sigma_1 + \Sigma_2)^{-1}(\mu_1 - \mu_2). \tag{D.5}$$

Thus, by (D.4) and (D.5), (D.3) becomes

$$\int N(x; \mu_1, \Sigma_1)N(x; \mu_2, \Sigma_2)dx = \frac{1}{(2\pi)^{n/2}|\Sigma_1 + \Sigma_2|^{1/2}} \exp\left(-\frac{1}{2}(\mu_1 - \mu_2)^\top(\Sigma_1 + \Sigma_2)^{-1}(\mu_1 - \mu_2)\right)$$
$$= N(\mu_1; \mu_2, \Sigma_1 + \Sigma_2).$$

This completes the proof. $\qquad\square$

## D.2 Sufficient Statistic for $\Omega$ in Corollary 3.3.1

**Lemma D.2.** Consider the hierarchical model where

$$\theta \mid \Omega, A, B \sim N\big(\mu(\Omega, A, B), , \Sigma(\Omega, A, B)\big),$$

with $\Omega$ as a parameter matrix, and $A$ and $B$ as fixed matrices. The mean $\mu(\Omega, A, B)$ and covariance $\Sigma(\Omega, A, B)$ depend on $(\Omega, A, B)$. Then, the conditional distribution $p(\theta|\Omega, A, B)$ can be expressed only in terms of $\Sigma(\Omega, A, B), A, B$ if and only if, for all pairs $(\Omega_1, \Omega_2)$ such that $\Sigma(\Omega_1, A, B) = \Sigma(\Omega_2, A, B)$, we also have $\mu(\Omega_1, A, B) = \mu(\Omega_2, A, B)$. That is,

$$\mu(\Omega, A, B) \text{ is determined by } \Sigma(\Omega, A, B), A, B$$
$$\Leftrightarrow p(\theta \mid \Omega, A, B) = p\big(\theta \mid \Sigma(\Omega, A, B), A, B\big).$$

*Proof.* If

$$\mu(\Omega_1, A, B) = \mu(\Omega_2, A, B),$$

whenever $\Sigma(\Omega_1, A, B) = \Sigma(\Omega_2, A, B)$, then for a given value of $\Sigma$, the pair $(\mu, \Sigma)$ does not depend on which $\Omega$ generated $\Sigma$. Hence specifying $\Sigma(\Omega, A, B), A, B$ alone suffices to determine the normal distribution of $\theta$. Thus $p(\theta \mid \Omega, A, B) = p(\theta \mid \Sigma(\Omega, A, B), A, B)$. Conversely, if

$$p(\theta \mid \Omega, A, B) = p\big(\theta \mid \Sigma(\Omega, A, B), A, B\big).$$

Then any two values $\Omega_1$ and $\Omega_2$ yielding the same $\Sigma(\Omega_1, A, B)$ and $\Sigma(\Omega_2, A, B)$ must produce the same distribution for $\theta$. Since $\theta$ is normally distributed, its mean must also match, i.e., $\mu(\Omega_1, A, B) = \mu(\Omega_2, A, B)$. This completes the proof. $\quad\square$

## D.3 Integral of Normal-Inverse-Wishart (NIW) distribution

**Lemma D.3.** Given a joint Normal-Inverse-Wishart (NIW) distribution
$$p(\theta, \Sigma') = NIW(\theta, \Sigma'; \mu', 1, \Psi', \nu'), \quad \theta \mid \Sigma' \sim N(\mu', \Sigma'), \quad \Sigma' \sim IW(\Psi', \nu'),$$
the marginal distribution of $\theta$ is a multivariate $t$-distribution:
$$\theta \sim t_{\nu'}\left(\mu', \frac{\Psi'}{\nu' - m + 1}\right),$$
with degrees of freedom $\nu'$, location parameter $\mu'$, and scale matrix $\frac{\Psi'}{\nu'-m+1}$.

*Proof.* To derive the marginal distribution of $\theta$, we integrate out $\Sigma'$:
$$p(\theta) = \int p(\theta \mid \Sigma')p(\Sigma')\, d\Sigma'. \tag{D.6}$$

The conditional distribution $p(\theta \mid \Sigma')$ is:
$$p(\theta \mid \Sigma') = \frac{1}{(2\pi)^{m/2}|\Sigma'|^{1/2}} \exp\left(-\frac{1}{2}(\theta - \mu')^\top (\Sigma')^{-1}(\theta - \mu')\right). \tag{D.7}$$

The marginal distribution $p(\Sigma')$ is:
$$p(\Sigma') = \frac{|\Psi'|^{\nu'/2}}{2^{\nu'm/2}\Gamma_m(\nu'/2)}|\Sigma'|^{-(\nu'+m+1)/2}\exp\left(-\frac{1}{2}\mathrm{tr}(\Psi'(\Sigma')^{-1})\right). \tag{D.8}$$

Substituting (D.7) and (D.8) into (D.6):
$$p(\theta) \propto \int |\Sigma'|^{-(\nu'+m+2)/2}\exp\left(-\frac{1}{2}\left[(\theta - \mu')^\top (\Sigma')^{-1}(\theta - \mu') + \mathrm{tr}(\Psi'(\Sigma')^{-1})\right]\right)d\Sigma'.$$

Let $S = (\theta - \mu')(\theta - \mu')^\top + \Psi'$, then:
$$p(\theta) \propto \int |\Sigma'|^{-(\nu'+m+2)/2}\exp\left(-\frac{1}{2}\mathrm{tr}\left((\Sigma')^{-1}S\right)\right)d\Sigma'. \tag{D.9}$$

Using the matrix integral identity ((Muirhead, 2009, Chapter 7.2) or (Gupta & Nagar, 2018, Chapter 1.4)):
$$\int |\Sigma|^{-(a+m+1)/2}\exp\left(-\frac{1}{2}\mathrm{tr}(\Sigma^{-1}B)\right)d\Sigma \propto |B|^{-a/2},$$
we identify $a = \nu' + 1$ and $B = S$. (D.9) becomes:
$$p(\theta) \propto |S|^{-(\nu'+1)/2}. \tag{D.10}$$

Expand $|S|$ using the matrix determinant lemma:
$$|S| = |\Psi'|\left(1 + (\theta - \mu')^\top (\Psi')^{-1}(\theta - \mu')\right). \tag{D.11}$$

Substituting (D.11) back to (D.10), we have
$$p(\theta) \propto \left(1 + \frac{(\theta - \mu')^\top (\Psi')^{-1}(\theta - \mu')}{\nu}\right)^{-(\nu+m)/2},$$
where $\nu = \nu' - m + 1$. This matches the kernel of a multivariate $t$-distribution:
$$\theta \sim t_\nu\left(\mu', \frac{\Psi'}{\nu}\right) = t_{\nu'}\left(\mu', \frac{\Psi'}{\nu' - m + 1}\right).$$
This completes the proof. □

### D.4 Regression Estimators

**Lemma D.4** (Estimation of $\alpha$, $\beta$ under Homoscedastic and Correlated Errors). Consider a multivariate linear regression model:

$$r = \alpha + F\beta + \epsilon^F, \quad \epsilon^F \sim N(0, \Omega^F), \tag{D.12}$$

where $\Omega^F$ is a constant positive definite covariance matrix across observations (i.e. the error term $\epsilon^F$ is homoscedastic but potentially correlated). Given observations $\{(r_l, F_l)\}_{l=1}^n$, the maximum likelihood estimators (MLE) for $(\alpha, \beta)$ are:

$$\widehat{\alpha} = \bar{r} - \overline{F}\widehat{\beta}, \quad \widehat{\beta} = \Big( \sum_{l=1}^n \widetilde{F}_l^\top (\widehat{\Omega}^F)^{-1} \widetilde{F}_l \Big)^{-1} \sum_{l=1}^n \widetilde{F}_l^\top (\widehat{\Omega}^F)^{-1} \widetilde{r}_l,$$

where

$$\bar{r} = \frac{1}{n} \sum_{l=1}^n r_l, \quad \widetilde{r}_l = r_l - \bar{r}, \quad \overline{F} = \frac{1}{n} \sum_{l=1}^n F_l, \quad \widetilde{F}_l = F_l - \overline{F}.$$

*Proof.* The log-likelihood function for the model is:

$$\log L(\alpha, \beta) = -\frac{n}{2} \log |\widehat{\Omega}^F| - \frac{1}{2} \sum_{l=1}^n (r_l - \alpha - F_l\beta)^\top (\widehat{\Omega}^F)^{-1} (r_l - \alpha - F_l\beta) + \text{const.} \tag{D.13}$$

Differentiate the log-likelihood (D.13) with respect to $\alpha$ and set it to zero:

$$\frac{\partial}{\partial \alpha} \log L(\alpha, \beta) = -\frac{1}{2} \sum_{l=1}^n \frac{\partial}{\partial \alpha} \Big[ (r_l - \alpha - F_l\beta)^\top (\widehat{\Omega}^F)^{-1} (r_l - \alpha - F_l\beta) \Big]$$

$$= \sum_{l=1}^n (\widehat{\Omega}^F)^{-1} (r_l - \alpha - F_l\beta) = 0.$$

Summing over all observations:

$$\sum_{l=1}^n (\widehat{\Omega}^F)^{-1} r_l - n(\widehat{\Omega}^F)^{-1}\alpha - \sum_{l=1}^n (\widehat{\Omega}^F)^{-1} F_l\beta = 0.$$

Solving for $\widehat{\alpha}$:

$$\widehat{\alpha} = \bar{r} - \overline{F}\widehat{\beta}. \tag{D.14}$$

Differentiate the log-likelihood (D.13) with respect to $\beta$ and set it to zero:

$$\frac{\partial}{\partial \beta} \log L(\alpha, \beta) = -\frac{1}{2} \sum_{l=1}^n \frac{\partial}{\partial \beta} \Big[ (r_l - \alpha - F_l\beta)^\top (\widehat{\Omega}^F)^{-1} (r_l - \alpha - F_l\beta) \Big]$$

$$= \sum_{l=1}^n F_l^\top (\widehat{\Omega}^F)^{-1} (r_l - \alpha - F_l\beta) = 0. \tag{D.15}$$

Using the expression for $\widehat{\alpha}$ (D.14), we have

$$r_l - \widehat{\alpha} - F_l\beta = r_l - (\bar{r} - \overline{F}\widehat{\beta}) - F_l\beta = (r_l - \bar{r}) - (F_l - \overline{F})\beta = \widetilde{r}_l - \widetilde{F}_l\beta.$$

Substituting back to (D.15) with $\alpha = \widehat{\alpha}$:

$$\sum_{l=1}^n F_l^\top (\widehat{\Omega}^F)^{-1} (\widetilde{r}_l - \widetilde{F}_l\beta) = 0.$$

Since $\sum_{l=1}^n \widetilde{r}_l = 0$ and $\sum_{l=1}^n \widetilde{F}_l = 0$, we have:

$$\sum_{l=1}^n \widetilde{F}_l^\top (\widehat{\Omega}^F)^{-1} \widetilde{r}_l - \sum_{l=1}^n \widetilde{F}_l^\top (\widehat{\Omega}^F)^{-1} \widetilde{F}_l\beta = 0.$$

Solving for $\widehat{\beta}$:

$$\widehat{\beta} = \left( \sum_{l=1}^{n} \widetilde{F}_l^{\top} (\widehat{\Omega}^F)^{-1} \widetilde{F}_l \right)^{-1} \sum_{l=1}^{n} \widetilde{F}_l^{\top} (\widehat{\Omega}^F)^{-1} \widetilde{r}_l.$$

Since $\epsilon^F$ is neither uncorrelated nor homoscedastic, remaining $\widehat{\Omega}^F$ is essential to obtain unbiased estimates of $\beta$.

This completes the proof. $\qquad\square$

# E   Proofs of Main Text

## E.1   Proof of Lemma 2.1

We separate the proof into two parts. One for (2.4) and another for (2.5):

*Proof of* (2.4).

$$p(\theta|q, \Omega) \propto L(\theta|q, \Omega)\pi(\theta)$$
$$= \exp\left\{ -\frac{1}{2}(q - P\theta)^{\mathsf{T}}\Omega^{-1}(q - P\theta) \right\} \exp\left\{ -\frac{1}{2}(\theta - \theta_0)^{\mathsf{T}}\Sigma_0^{-1}(\theta - \theta_0) \right\} \qquad \text{(By (2.2) and (2.3))}$$
$$= \exp\left\{ -\frac{1}{2} \left( \theta^{\mathsf{T}}P^{\mathsf{T}}\Omega^{-1}P\theta - 2q^{\mathsf{T}}\Omega^{-1}P\theta + q^{\mathsf{T}}\Omega^{-1}q + \theta^{\mathsf{T}}\Sigma_0^{-1}\theta - 2\theta_0^{\mathsf{T}}\Sigma_0^{-1}\theta + \theta_0^{\mathsf{T}}\Sigma_0^{-1}\theta_0 \right) \right\}$$
$$= \exp\left\{ -\frac{1}{2} \left[ \theta^{\mathsf{T}}(P^{\mathsf{T}}\Omega^{-1}P + \Sigma_0^{-1})\theta - 2(q^{\mathsf{T}}\Omega^{-1}P + \theta_0^{\mathsf{T}}\Sigma_0^{-1})\theta + q^{\mathsf{T}}\Omega^{-1}q + \theta_0^{\mathsf{T}}\Sigma_0^{-1}\theta_0 \right] \right\}, \qquad \text{(E.1)}$$

where the third equation is the result of the symmetric property of $\Sigma_0$ and $\Omega$.

To simplify the above expression, we introduce

$$G := \Sigma_0^{-1} + P^{\mathsf{T}}\Omega^{-1}P,$$
$$D := \Sigma_0^{-1}\theta_0 + P^{\mathsf{T}}\Omega^{-1}q,$$
$$A := \theta_0^{\mathsf{T}}\Sigma_0^{-1}\theta_0 + q^{\mathsf{T}}\Omega^{-1}q,$$

then, we have:

$$\theta^{\mathsf{T}}G\theta - 2D^{\mathsf{T}}\theta + A = (G\theta)^{\mathsf{T}}G^{-1}G\theta - 2D^{\mathsf{T}}G^{-1}G\theta + A$$
$$= (G\theta - D)^{\mathsf{T}}G^{-1}(G\theta - D) + A - D^{\mathsf{T}}G^{-1}D$$
$$= (\theta - G^{-1}D)^{\mathsf{T}}G(\theta - G^{-1}D) + A - D^{\mathsf{T}}G^{-1}D.$$

Therefore, (E.1) becomes

$$p(\theta|q, \Omega) \propto \exp\left\{ -\frac{1}{2}(\theta - G^{-1}D)^{\mathsf{T}}G(\theta - G^{-1}D) \right\} = N(\theta; G^{-1}D, G^{-1}).$$

This completes the proof. $\qquad\square$

*Proof of* (2.5). From (2.1), we have:

$$p(r|\theta) = N(r; \theta, \Sigma). \qquad \text{(E.2)}$$

From Lemma C.1, we have:

$$\theta_0 = \delta(\Sigma + \Sigma_0)w_{\text{cap}}. \qquad \text{(E.3)}$$

From (2.4), we have:

$$p(\theta|q) = N\left(\theta; G^{-1}\left(\Sigma_0^{-1}\theta_0 + P^{\mathsf{T}}\Omega^{-1}q\right), G^{-1}\right). \qquad \text{(E.4)}$$

Thus,

$$\widetilde{r} \sim p(r|q) = \int p(\widetilde{r}|\theta)p(\theta|q)d\theta$$

$$= \int N\left(\widetilde{r}; \theta, \Sigma\right) N\left(\theta; G^{-1}(\Sigma_0^{-1}\theta_0 + P^{\mathsf{T}}\Omega^{-1}q), G^{-1}\right) d\theta \qquad \text{(By (E.2) and (E.4))}$$

$$= N\left(\widetilde{r}; G^{-1}(\Sigma_0^{-1}\theta_0 + P^{\mathsf{T}}\Omega^{-1}q), \Sigma + G^{-1}\right) \qquad \text{(From Lemma D.1)}$$

$$= N\left(\widetilde{r}; G^{-1}\left[\delta(\Sigma_0^{-1}\Sigma + I)w_{\text{cap}} + P^{\mathsf{T}}\Omega^{-1}q\right], \Sigma + G^{-1}\right). \qquad \text{(By (E.3))}$$

This completes the proof. $\qquad\qquad\qquad\square$

## E.2  Proof of Theorem 3.1

*Proof of Theorem 3.1.* The posterior $\theta$ given the data is proportional to the product of three Gaussian densities:

$$p(\theta|q, \Omega, F, \Omega^F) \propto \underbrace{p(q|\theta, F, \Omega)}_{\text{Observation Likelihood}} \cdot \underbrace{p(\theta|F, \Omega^F)}_{\text{Features Likelihood}} \cdot \underbrace{\pi(\theta)}_{\text{Prior}}, \qquad (\text{E.5})$$

where the observation likelihood, features likelihood, and prior distribution are respectively:

$$p(q|\theta, F, \Omega) = \exp\left(-\frac{1}{2}\left[q - P(\alpha + F\beta + \gamma\theta)\right]^{\mathsf{T}}\Omega^{-1}\left[q - P(\alpha + F\beta + \gamma\theta)\right]\right), \qquad (\text{By (3.4)})$$

$$p(\theta|F, \Omega^F) = \exp\left(-\frac{1}{2}\left[\theta - (\alpha^F + F\beta^F)\right]^{\mathsf{T}}(\Omega^F)^{-1}\left[\theta - (\alpha^F + F\beta^F)\right]\right), \qquad (\text{By (3.3)})$$

$$\pi(\theta) = \exp\left(-\frac{1}{2}(\theta - \theta_0)^{\mathsf{T}}\Sigma_0^{-1}(\theta - \theta_0)\right). \qquad (\text{By (3.2)})$$

Combining all quadratic forms in the exponent (ignoring constants) of $p(\theta|q, \Omega, F, \Omega^F)$, we have

$$-\frac{1}{2}\Big[(\theta - \theta_0)^{\mathsf{T}}\Sigma_0^{-1}(\theta - \theta_0) + [\theta - (\alpha^F + F\beta^F)]^{\mathsf{T}}(\Omega^F)^{-1}[\theta - (\alpha^F + F\beta^F)]$$

$$+ [q - P\alpha - PF\beta - \gamma P\theta]^{\mathsf{T}}\Omega^{-1}[q - P\alpha - PF\beta - \gamma P\theta]\Big]$$

$$= -\frac{1}{2}\Big[\theta^{\mathsf{T}}\Sigma_0^{-1}\theta - 2\theta_0^{\mathsf{T}}\Sigma_0^{-1}\theta + \theta_0^{\mathsf{T}}\Sigma_0^{-1}\theta_0$$

$$+ \theta^{\mathsf{T}}(\Omega^F)^{-1}\theta - 2(\alpha^F + F\beta^F)^{\mathsf{T}}(\Omega^F)^{-1}\theta + (\alpha^F + F\beta^F)^{\mathsf{T}}(\Omega^F)^{-1}(\alpha^F + F\beta^F)$$

$$+ \gamma^2\theta^{\mathsf{T}}P^{\mathsf{T}}\Omega^{-1}P\theta - 2\gamma(q - P\alpha - PF\beta)^{\mathsf{T}}\Omega^{-1}P\theta$$

$$+ (q - P\alpha - PF\beta)^{\mathsf{T}}\Omega^{-1}(q - P\alpha - PF\beta)\Big]$$

$$= -\frac{1}{2}\Big[\theta^{\mathsf{T}}\underbrace{\left[\Sigma_0^{-1} + (\Omega^F)^{-1} + \gamma^2 P^{\mathsf{T}}\Omega^{-1}P\right]}_{G^M}\theta$$

$$- 2\theta^{\mathsf{T}}\underbrace{\left[\Sigma_0^{-1}\theta_0 + (\Omega^F)^{-1}(\alpha^F + F\beta^F) + \gamma P^{\mathsf{T}}\Omega^{-1}(q - P\alpha - PF\beta)\right]}_{b} + \text{constant}\Big],$$

where the second step groups similar terms and the first step expands the quadratic form.

Completing the square in the form

$$-\frac{1}{2}\left(\theta - \mu_{\theta|q,\Omega,F,\Omega^F}\right)^{\mathsf{T}}G^M\left(\theta - \mu_{\theta|q,\Omega,F,\Omega^F}\right),$$

the posterior distribution (E.5) becomes:

$$p(\theta|q, \Omega, F, \Omega^F) = N\left(\theta; \mu_{\theta|q,\Omega,F,\Omega^F}, (G^M)^{-1}\right)$$

where

$$G^M := \Sigma_0^{-1} + (\Omega^F)^{-1} + \gamma^2 P^{\mathsf{T}}\Omega^{-1}P,$$

$$\mu_{\theta|q,\Omega,F,\Omega^F} := (G^M)^{-1}b$$

$$= (G^M)^{-1}\left[\Sigma_0^{-1}\theta_0 + (\Omega^F)^{-1}(\alpha^F + F\beta^F) + \gamma P^{\mathsf{T}}\Omega^{-1}(q - P\alpha - PF\beta)\right]$$

This completes the proof. $\qquad\qquad\qquad\square$

### E.3 Proof of Theorem 3.2

*Proof of Theorem 3.2.* From the $\theta \leftrightarrow F$ model (Definition 3.1),

$$L(\theta|F, \Omega^F) = p(F, \Omega^F|\theta) \propto \exp\Big(-\frac{1}{2}[F\beta^F - (\theta - \alpha^F)]^\mathsf{T}(\Omega^F)^{-1}[F\beta^F - (\theta - \alpha^F)]\Big).$$

Thus, we have

$$p(\theta|F, \Omega^F) \propto L(\theta|F, \Omega^F)\pi(\theta)$$

$$\propto \exp\Big(-\frac{1}{2}[F\beta^F - (\theta - \alpha^F)]^\mathsf{T}(\Omega^F)^{-1}[F\beta^F - (\theta - \alpha^F)]\Big) \exp\Big(-\frac{1}{2}(\theta - \theta_0)^\mathsf{T}\Sigma_0^{-1}(\theta - \theta_0)\Big)$$

$$\text{(By (3.6) and (3.5))}$$

$$\propto \exp\Big(-\frac{1}{2}\big[(\theta - \alpha^F - F\beta^F)^\mathsf{T}(\Omega^F)^{-1}(\theta - \alpha^F - F\beta^F) + (\theta - \theta_0)^\mathsf{T}\Sigma_0^{-1}(\theta - \theta_0)\big]\Big).$$

$$\propto \exp\Big(-\frac{1}{2}\big[\theta^\mathsf{T}[(\Omega^F)^{-1} + \Sigma_0^{-1}]\theta - 2[\Sigma_0^{-1}\theta_0 + (\Omega^F)^{-1}(\alpha^F + F\beta^F)]^\mathsf{T}\theta + \text{const}\big]\Big)$$

$$\propto \exp\Big(-\frac{1}{2}\big[\theta^\mathsf{T}G^F\theta - 2(D^F)^\mathsf{T}\theta + \text{const}\big]\Big),$$

$$\propto \exp\Big(-\frac{1}{2}\big[(\theta - \mu_{\theta|F,\Omega^F})^\mathsf{T}G^F(\theta - \mu_{\theta|F,\Omega^F}) - \mu_{\theta|F,\Omega^F}^\mathsf{T}G^F\mu_{\theta|F,\Omega^F} + \text{const}\big]\Big), \tag{E.6}$$

where $G^F := (\Omega^F)^{-1} + \Sigma_0^{-1}$ and $D^F := \Sigma_0^{-1}\theta_0 + (\Omega^F)^{-1}(\alpha^F + F\beta^F)$ and

$$\mu_{\theta|F,\Omega^F} = (G^F)^{-1}D^F = (G^F)^{-1}\big[\Sigma_0^{-1}\theta_0 + (\Omega^F)^{-1}(\alpha^F + F\beta^F)\big].$$

Substituting back to (E.6), the posterior distribution is:

$$p(\theta|F, \Omega^F) = N\big(\theta; (G^F)^{-1}\big[\Sigma_0^{-1}\theta_0 + (\Omega^F)^{-1}(\alpha^F + F\beta^F)\big], (G^F)^{-1}\big).$$

This completes the proof. $\qquad\square$

### E.4 Proof of Theorem 3.3

*Proof of Theorem 3.3.* The first two relations (3.7) and (3.8), from (2.4), leads to:

$$p(\theta|q, \Omega) = N\big(\theta; G^{-1}\big(\Sigma_0^{-1}\theta_0 + P^\mathsf{T}\Omega^{-1}q\big), G^{-1}\big), \tag{E.7}$$

where $G := \Sigma_0^{-1} + P^\mathsf{T}\Omega^{-1}P$. The third relation (3.9) leads to

$$p(q|F, \Omega^F) = N(q; P(\alpha + F\beta), P\Omega^F P^\mathsf{T}). \tag{E.8}$$

Then, the posterior mean distribution

$$p(\theta|F, \Omega^F) = \iint p(\theta, q, \Omega|F, \Omega^F)dqd\Omega,$$

$$= \int\Big(\int p(\theta, q|\Omega, F, \Omega^F)dq\Big)\pi(\Omega)d\Omega,$$

$$= \int\Big(\int p(\theta|q, \Omega)p(q|F, \Omega^F)dq\Big)\pi(\Omega)d\Omega, \tag{E.9}$$

where the last step follows the conditional independence between $\theta$ and $(F, \Omega^F)$ given $(q, \Omega)$. Also, it is clear to see that

$$\theta|\Omega, F, \Omega^F \sim \int p(\theta|q, \Omega)p(q|F, \Omega^F)dq. \tag{E.10}$$

Both $p(\theta|q, \Omega)$ (E.7) and $p(q|F, \Omega^F)$ (E.8) are Gaussian, so the inside integral of (E.9) is also Gaussian, with

$$\int p(\theta|q, \Omega)p(q|F, \Omega^F)dq = N\big(\theta; \mu_{\theta|\Omega,F,\Omega^F}, \Sigma_{\theta|\Omega,F,\Omega^F}\big).$$

By the law of total expectation,

$$\mu_{\theta|\Omega,F,\Omega^F} = \mathbb{E}[\theta|\Omega, F, \Omega^F]$$

$$= \int \mathbb{E}[\theta|q, \Omega] p(q|F, \Omega^F) dq \qquad\qquad (\text{By (E.10)})$$

$$= \int G^{-1} \left( \Sigma_0^{-1} \theta_0 + P^\mathsf{T} \Omega^{-1} q \right) N \left( q; P(\alpha + F\beta), P\Omega^F P^\mathsf{T} \right) dq \qquad (\text{By (E.7) and (E.8)})$$

$$= G^{-1} \left( \Sigma_0^{-1} \theta_0 + P^\mathsf{T} \Omega^{-1} P(\alpha + F\beta) \right). \qquad (\text{By } \int q \cdot p(q|F, \Omega^F) dq = \mathbb{E}[q|F, \Omega^F].)$$

Using the law of total variance,

$$\Sigma_{\theta|\Omega, F, \Omega^F} = \mathrm{Var}[\theta|\Omega, F, \Omega^F]$$

$$= \mathbb{E}\left[ \mathrm{Var}[\theta|q, \Omega, F, \Omega^F] \right] + \mathrm{Var}\left[ \mathbb{E}[\theta|q, \Omega, F, \Omega^F] \right] \qquad (\text{By the law of total variance})$$

$$= \mathbb{E}\left[ \mathrm{Var}[\theta|q, \Omega] \right] + \mathrm{Var}\left[ \mathbb{E}[\theta|q, \Omega] \right] \qquad (\text{By conditional independence})$$

$$= \int \underbrace{\mathrm{Var}[\theta|q, \Omega]}_{G^{-1}} p(q|F, \Omega^F) dq + \mathrm{Var}\left[ \underbrace{\mathbb{E}[\theta \mid q, \Omega]}_{G^{-1}\left[\Sigma_0^{-1}\theta_0 + P^\mathsf{T}\Omega^{-1}q\right]} \right] \qquad (\text{By (E.7)})$$

$$= G^{-1} + G^{-1} P^\mathsf{T} \Omega^{-1} \mathrm{Var}[q] \Omega^{-1} P G^{-1} \qquad (\text{By } \int p(q|F, \Omega^F) = 1 \text{ and } \mathrm{Var}[Mq] = M\mathrm{Var}[q]M^{-1})$$

$$= G^{-1} + G^{-1} P^\mathsf{T} \Omega^{-1} \left( P\Omega^F P^\mathsf{T} \right) \Omega^{-1} P G^{-1}. \qquad (\text{By (E.8)})$$

This completes the proof. $\qquad\square$

### E.5 Proof of Corollary 3.3.1

*Proof of Corollary 3.3.1.* From Lemma D.2, we have

$$\theta|\Sigma', F, \Omega^F \sim N\left( \mu', \Sigma' \right).$$

From (3.11), we have

$$\Sigma' \sim IW(\Psi', \nu')$$

We obtain the joint distribution by definition of the Normal-Inverse-Wishart distribution:

$$p(\theta, \Sigma'|F, \Omega^F) = p(\theta|\Sigma', F, \Omega^F)\pi(\Sigma')$$

$$= N\left( \mu', \Sigma' \right) IW(\Psi', \nu')$$

$$= NIW(\mu', 1, \Psi', \nu'). \qquad\qquad (\text{E.11})$$

Then the posterior mean distribution becomes

$$p(\theta|F, \Omega^F) = \int p(\theta, \Sigma'|F, \Omega^F) d\Sigma',$$

$$= \int NIW(\theta, \Sigma'; \mu', 1, \Psi', \nu') d\Sigma', \qquad (\text{By (E.11)})$$

$$= t_{\nu'}\left( \theta; \mu', \frac{\Psi'}{\nu' - m + 1} \right),$$

where the last step follows Lemma D.3. This completes the proof. $\qquad\square$

# F   Experimental Details

## F.1   Sharpe Ratio Maximization

In our experiment (Section 4), we consider a standardized version of the mean-variance optimization framework (Definition 2.1), taking Sharpe ratio as its maximization objective:

**Definition F.1** (Mean-Variance Optimization on Sharpe ratio). Let $r \in \mathbb{R}^m$ be the returns of the $m$ assets and $\widetilde{r}$ be its prediction. The optimization problem, under the constraint of (1) no leverage and (2) long only, is:

$$\max_{w} \left( \frac{w^T \mathbb{E}[\widetilde{r}]}{\sqrt{w^T \mathrm{Cov}[\widetilde{r}] w}} \right) \text{ s.t. } \sum_{i=1}^{n} w_i = 1 \text{ and } w_i \geq 0.$$

## F.2   Table of Indicators

| Indicator | Description | Hyperparameters |
|---|---|---|
| ATR | Measures market volatility based on price range. | Window length (default 14) |
| ADX | Measures the strength of a trend. | Window length (default 14) |
| EMA | Weighted moving average prioritizing recent prices. | Window length (default 14) |
| MACD | Difference between short- and long-term EMAs, indicates momentum shifts. | Fast EMA window length (12), Slow EMA window length (26), Signal window length (9) |
| SMA | Average of prices over a specified window length, indicating short-term trends. | Window length (default 20) |
| RSI | Momentum oscillator identifying overbought/oversold conditions. | Window length (default 14) |
| BB (Upper & Lower) | Measures price volatility, expanding during high volatility and contracting during low. | Window length (default 20), Standard deviation multiplier (default 2) |
| OBV (normalized) | Volume indicator combined with price, normalized to range [0, 1]. | None (computed from price and volume) |

Table 3: Common Indicators Used

## F.3   Tables of Datasets

Table 4: S&P Sector ETF Components

| ETF Ticker | Start Date | End Date |
|---|---|---|
| XLB | 2004-04-13 | 2024-02-22 |
| XLE | 2004-04-13 | 2024-02-22 |
| XLF | 2004-04-13 | 2024-02-22 |
| XLI | 2004-04-13 | 2024-02-22 |
| XLK | 2004-04-13 | 2024-02-22 |
| XLP | 2004-04-13 | 2024-02-22 |
| XLU | 2004-04-13 | 2024-02-22 |
| XLV | 2004-04-13 | 2024-02-22 |
| XLY | 2004-04-13 | 2024-02-22 |
| XLRE | 2015-10-08 | 2024-02-22 |
| XLC | 2018-06-19 | 2024-02-22 |

Table 5: DJIA Components

| Stock Ticker | Start Date | End Date |
|---|---|---|
| AA | 1994-01-05 | 2013-09-22 |
| AIG | 2004-04-08 | 2008-09-21 |
| AAPL | 2015-03-19 | 2024-02-22 |
| AMGN | 2020-08-31 | 2024-02-22 |
| AXP | 1994-01-05 | 2024-02-22 |
| BA | 1994-01-05 | 2024-02-22 |
| BAC | 2008-02-19 | 2013-09-22 |
| C | 1999-11-01 | 2009-06-07 |
| CAT | 1994-01-05 | 2024-02-22 |
| CSCO | 2009-06-08 | 2024-02-22 |
| CVX | 2008-02-19 | 2024-02-22 |
| DD | 1994-01-05 | 2019-04-01 |
| DIS | 1994-01-05 | 2024-02-22 |
| FL | 1994-01-05 | 1997-03-17 |
| GE | 1994-01-05 | 2018-06-25 |
| GS | 2013-09-23 | 2024-02-22 |
| GT | 1994-01-05 | 1999-11-01 |
| HD | 1999-11-01 | 2024-02-22 |
| HON | 1994-01-05 | 2024-02-22 |
| HPQ | 1997-03-17 | 2013-09-22 |
| IBM | 1994-01-05 | 2024-02-22 |
| INTC | 1999-11-01 | 2024-02-22 |
| IP | 1994-01-05 | 2004-04-08 |
| JNJ | 1997-03-17 | 2024-02-22 |
| JPM | 1994-01-05 | 2024-02-22 |
| KO | 1994-01-05 | 2024-02-22 |
| MCD | 1994-01-05 | 2024-02-22 |
| MMM | 1994-01-05 | 2024-02-22 |
| MO | 1994-01-05 | 2008-02-18 |
| MRK | 1994-01-05 | 2024-02-22 |
| MSFT | 1999-11-01 | 2024-02-22 |
| NKE | 2013-09-23 | 2024-02-22 |
| PFE | 2004-04-08 | 2024-02-22 |
| PG | 1994-01-05 | 2024-02-22 |
| T | 1994-01-05 | 2015-03-18 |
| TRV | 1997-03-17 | 2024-02-22 |
| UNH | 2012-09-24 | 2024-02-22 |
| VZ | 2004-04-08 | 2024-02-22 |
| WBA | 2018-06-26 | 2024-02-22 |
| WMT | 1994-01-05 | 2024-02-22 |
| XOM | 1994-01-05 | 2024-02-22 |

## F.4 Figures of Asset Allocation

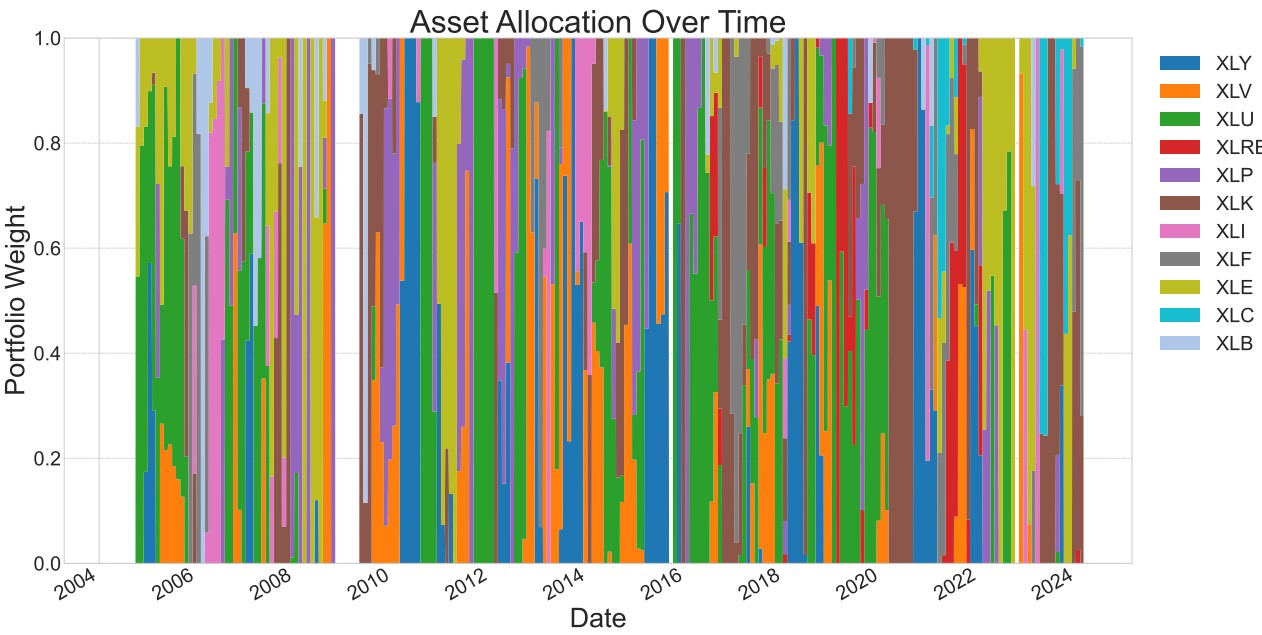

Figure 6: Asset Allocation of traditional MV model (rolling window: 100 days) on SPDR Sectors ETFs Dataset.

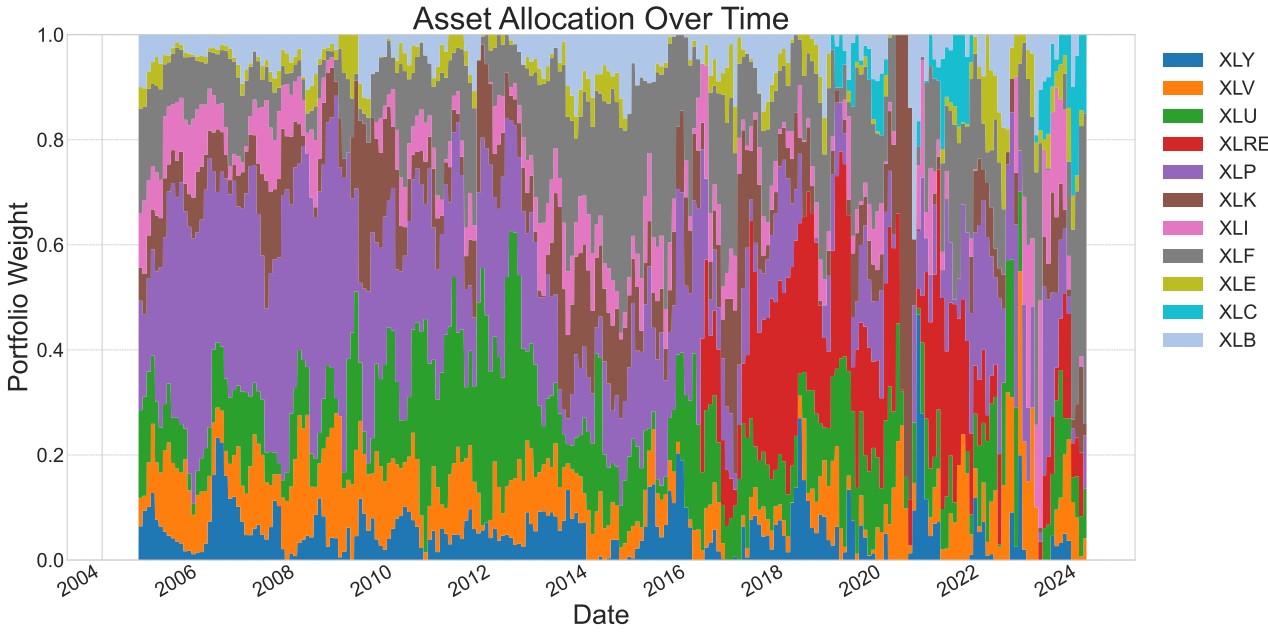

Figure 7: Asset Allocation of SLP-BL model (rolling window: 100 days) on SPDR Sectors ETFs Dataset.

## F.5 Turnover Rate Analysis

| | Average Turnover Rate (%) on Dow Jones Index Dataset | Average Turnover Rate (%) on SPDR Sector ETFs Dataset |
|---|---|---|
| MV (50d) | 65.48 | 61.75 |
| BL (50d) | 34.85 | 24.30 |
| MV (80d) | 53.94 | 49.38 |
| BL (80d) | 25.62 | 19.56 |
| MV (100d) | 47.79 | 48.73 |
| BL (100d) | 23.41 | 18.63 |
| MV (120d) | 44.30 | 41.72 |
| BL (120d) | 20.89 | 18.09 |
| MV (150d) | 39.70 | 38.84 |
| BL (150d) | 19.17 | 17.00 |

Table 6: Average Turnover Rate (%) for SLP-BL and Markowitz model on the Dow Jones Index and SPDR Sector ETFs Datasets.

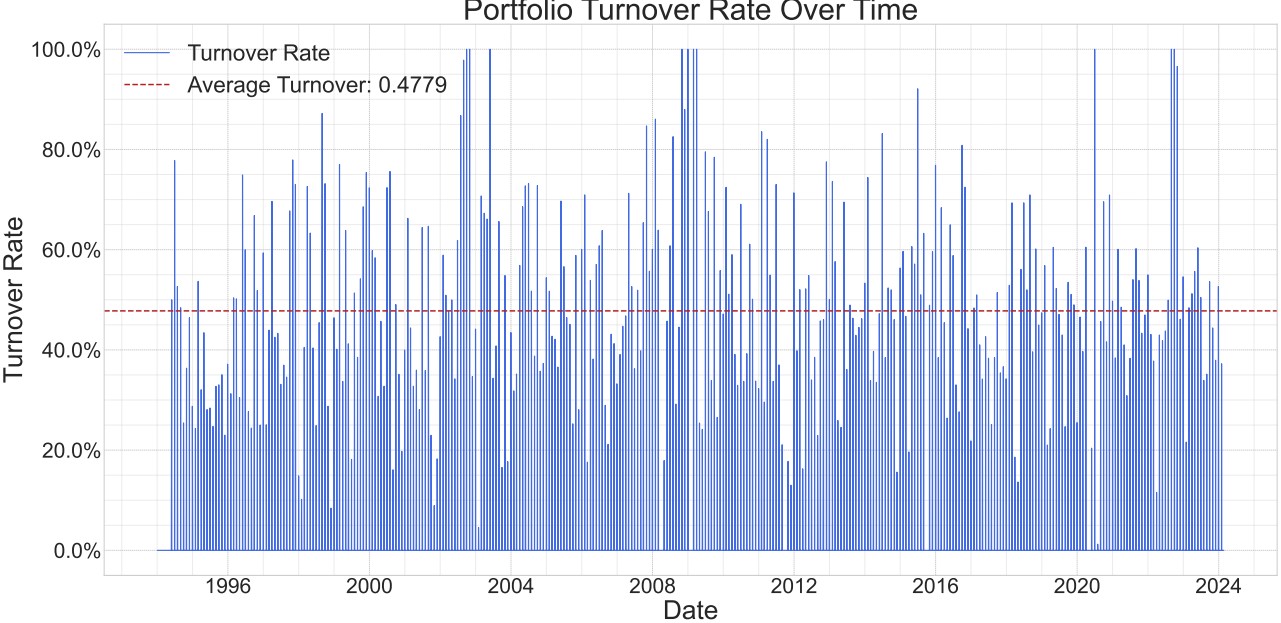

Figure 8: Turnover rate of traditional MV model (rolling window: 100 days) on Dow Jones Index Dataset.

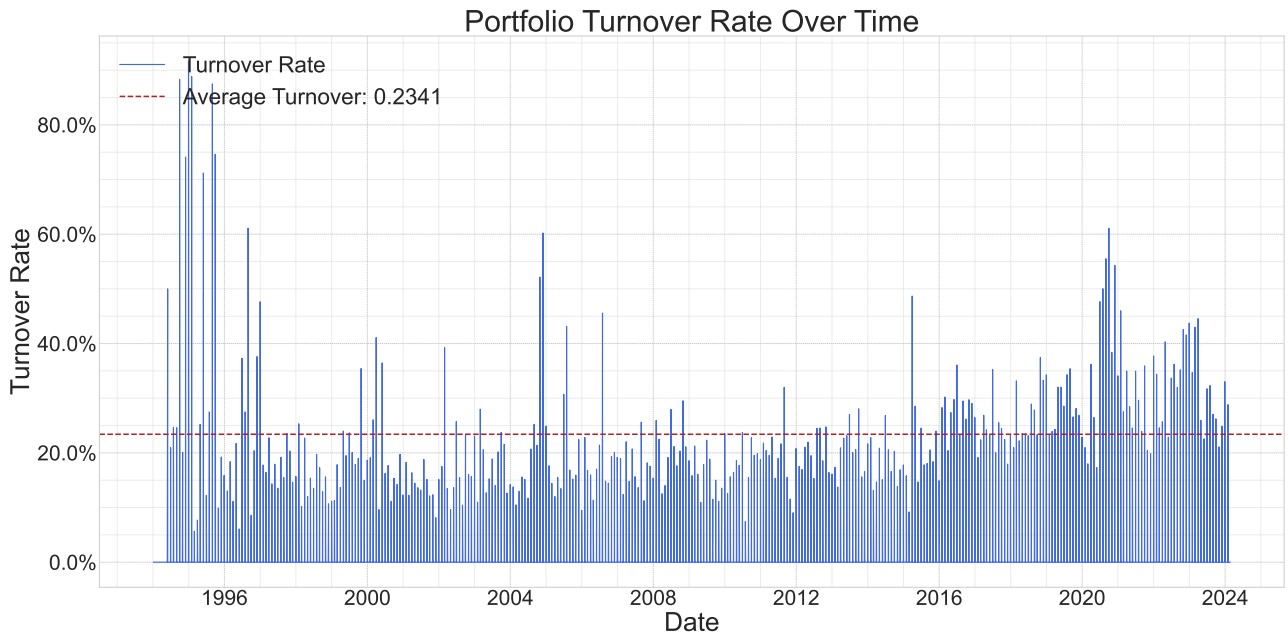

Figure 9: Turnover rate of SLP-BL model (rolling window: 100 days) on Dow Jones Index Dataset.

