# OpenReview forum: "Latent Variable Estimation in Bayesian Black-Litterman Models"
_ICML.cc/2025/Conference — ICML 2025 poster_

### Official Review · Reviewer_KJCs · 2025-02-27

**Overall Recommendation:** 4

**Summary:**

The paper proposes Bayesian models for Portfolio management with good theoretical results and empirical validation. From what I can understand the contribution of the work is computational efficiency of portfolio management. Could the code be provided to do this?

Given no code, I see no academic value for this work.

The empirical validation is far too small to allow for any reasonable examination of the academic value of this work.

## update after rebuttal
I have raised my score due to code release.

**Claims And Evidence:**

The claims and evidence are sufficient.

**Essential References Not Discussed:**

N/A

**Experimental Designs Or Analyses:**

The experimental design seems sufficient for the paper.

**Methods And Evaluation Criteria:**

The methods and evaluation are sufficient.

**Other Comments Or Suggestions:**

I'm not sure how or why this work is relevant or useful to the ICML community or the machine learning community.

**Other Strengths And Weaknesses:**

N/A

**Questions For Authors:**

Please answer how this is relevant to the ICML community.

**Relation To Broader Scientific Literature:**

N/A

**Theoretical Claims:**

There is no justification given as to why Bayesian comes up in portfolio management.

---

> ### Author Rebuttal · Authors · 2025-04-01
>
> >**Reviewer's Comment:** Could the code be provided to do this? Given no code, I see no academic value for this work.
>
> We provide code for reproducibility and the latest paper revision in this **[anonymous dropbox folder](https://www.dropbox.com/scl/fo/i13bhu138gjk76cf5r44v/ACog6LpDbYdBQbB87jtJA-Q?rlkey=mpv3b4xgbmr1cohaen6roxoyf&st=zndq6hnx&dl=0).**
>
> **On Code Availability.** We originally withheld the code due to industrial IP auditing procedures. However, per the reviewer's request, we have now completed the necessary internal reviews and are able to share an anonymized version of the code for reproducibility purposes.
>
> >**Reviewer's Comment:** The empirical validation is far too small to allow for any reasonable examination of the academic value of this work.
>
> We have extended the backtest period of the experiment on the DJI dataset from 20 years to 30 years, as shown by Table 2, Section 4 and Figure 5, Appendix G.4. We have also added a turnover rate analysis (Appendix G.6) to further demonstrate our model outperformance.
>
> We note that our experiments aim to prove the concept—specifically, to prove that this machine learning model could be readily implemented and useful (compared to the benchmarks). The two datasets we choose are practically useful as they provide some benefits compared to a larger set of stocks: ease of trading, smaller impact of slippage (due to higher liquidity), fewer trades per period (thus lower transaction costs), lower managerial costs, and lower computational costs.
>
> We need to clarify that the backtest period should be carefully chosen to avoid unfair comparison. Specifically, although the DJIA covers a long history, some stock data might be inaccessible. Omitting them (as many portfolio studies do) could introduce selection bias because delisted stocks often underperform. Thus, we believe our experimental results now (20 yrs for sector ETFs; 30 yrs on the Dow Jones Index) are sufficient for showing the outperformance of our models.
>
> >**Reviewer's Comment:** There is no justification given as to why Bayesian comes up in portfolio management.
>
> As we mention in \textbf{Why Bayesian? } in the related works (Appendix C), Bayesian framework has long been advocated and used in portfolio management. We state:
>
> *“To address the parameter estimation risk in traditional portfolio optimization shown by (Markowitz, 1952; Kalymon, 1971), Barry (1974); Klein & Bawa (1976); Brown (1976) advocate Bayesian framework upon prior information in portfolio optimization.”*
>
> Moreover, it is proved that the Bayesian framework has several benefits, as we state:
>
> *“Foundational works by (Jorion, 1986) and (Black & Litterman, 1992) demonstrate how Bayesian shrinkage improves covariance estimation, reducing overfitting and highly sensitive weight in Markowitz-style allocations (Meucci, 2005; DeMiguel et al., 2009).”*
>
> We hope this background information clarifies the use of Bayesian.
>
> >**Reviewer's Comment:** Please answer how this is relevant to the ICML community.
>
> From an essential and practical perspective, our work aims to take different types of data under different scenarios — some data is given, while others are not — to make predictions and decisions. Thus, it addresses machine learning problems using machine learning methods, and should be considered relevant to general ICML community.
> Specifically,
> 1. **Different types of data** include: raw data, feature data extracted from raw data, feature data involving additional information, and heuristic expert knowledge (views). One of the major focuses in our model design is to capture the effects of these data through Bayesian networks with latent variables, a common methodology in machine learning research. These data are described in the introduction (page 1), Sec. 3.2, 3.3, and 3.4 (page 3-5).
> 2. **Different scenarios** refer to Sec. 3.3, where investor views are observed, and Sec. 3.4, where no subjective views are given. In the scenario of Sec. 3.4, our model is showcased by two configurations for handling different types of data. These scenarios are described in the paragraph before Sec. 3.1 (page 3) and the paragraph after remark. 3.3 (page 5).
> 3. **Predictions** refer to the posterior probability distribution of unobserved asset returns given data estimated by each designed model. The prediction problems are described in problems 1, 2, 3 (page 2, 4, 5). The prediction models are Def 2.2, 3.3, 3.4, and 3.6, and they make predictions in Lemma 2.1, Corollary 3.1.1, 3.2.1, and 3.3.1, correspondingly.
> 4. **Decisions** refer to the solution of the portfolio optimization problem (Def. 2.1, page 2) based on the estimations of each designed model, including Lemma 2.1, Thm 3.1, 3.2, and 3.3. The portfolio optimization problem is also a popular topic in machine learning research.
> In this perspective, we deem our work to fall under the scope of machine learning and satisfy the topics of interest (https://icml.cc/Conferences/2025/CallForPapers) of ICML.

---

> > ### Comment · Reviewer_KJCs · 2025-04-05
> >
> > I have raised my score given the code release.

---

> > > ### Author Response · Authors · 2025-04-06
> > >
> > > We are very happy that our revisions and clarifications have met your expectations. Thank you again for your detailed review! Your constructive comments have greatly improved this draft.

---

### Official Review · Reviewer_fEJh · 2025-03-05

**Overall Recommendation:** 2

**Summary:**

The paper extends the classical Black-Litterman model by incorporating asset features. In the traditional model, investor views and their associated uncertainty are assumed to be given. The author proposes leveraging asset features to estimate both investor views and their uncertainty. Two models are introduced: the first assumes that asset features influence both investor views and uncertainty through a shared hyperparameter, while the second allows asset features to directly influence investor views. Numerical experiments demonstrate that the proposed approach outperforms the Markowitz and market-index baselines.

**Claims And Evidence:**

The paper is well written and use abundant figures to demonstrate the proposed graphical models.

**Essential References Not Discussed:**

The paper cites sufficient papers in the literature.

**Experimental Designs Or Analyses:**

- **Features Used in Experiments:** What asset features are used in the experiment section? Since the proposed method requires additional input (asset features) compared to the baseline methods, one could easily construct a portfolio that outperforms the baselines by simply assigning greater weight to the tech sector.

- **Choice Between Configurations:** Can the authors provide a discussion on how to choose between the two proposed configurations? Since, in practice, practitioners primarily need a formula to compute portfolio weights, such a discussion would be valuable. Similarly, what kinds of asset features are most suitable for the proposed method?

**Methods And Evaluation Criteria:**

Yes. The metrics used in the paper for porfolio selection are standard.

**Other Comments Or Suggestions:**

See Experimental Designs Or Analyses.

**Other Strengths And Weaknesses:**

Strengths

 The paper proposes a Bayesian framework for incorporating asset features, and the resulting formulas have closed-form expressions. This is advantageous and practical, as it allows for easy computation.

**Questions For Authors:**

See Experimental Designs Or Analyses.

**Relation To Broader Scientific Literature:**

The paper extends Black-Litterman model and is related to that literature.

**Theoretical Claims:**

I checked some of the proofs and did not find obvious errors.

---

> ### Author Rebuttal · Authors · 2025-04-01
>
> Thanks for the reviews.
> The latest revision is readily available in the **[anonymous dropbox folder](https://www.dropbox.com/scl/fo/i13bhu138gjk76cf5r44v/ACog6LpDbYdBQbB87jtJA-Q?rlkey=mpv3b4xgbmr1cohaen6roxoyf&st=zndq6hnx&dl=0).**
> Any changes made from the submitted version are highlighted in blue in this updated draft.
>
> ## Features Used in Experiments.
> >**Reviewer's Comment:** What asset features are used in the experiment section?
>
> The asset features we use in our experiments are listed in Appendix G.3. They include common indicators (e.g., EMA, MACD, RSI, ...) in financial analysis. The idea behind selecting these features is to keep things generic, following configuration 1 (as explained in Remark 3.3). We state the description of our features usage in **Backtest Task. (page 8)** with:
> *“In the model, the prior is set as traditional Markowitz model and the features are selected based on nine generic indicators (Table 5) derived from asset-specific data.”*
>
> >**Reviewer's Comment:** Since the proposed method requires additional input (asset features) compared to the baseline methods, one could easily construct a portfolio that outperforms the baselines by simply assigning greater weight to the tech sector.
>
> We would like to clarify that our method does not necessarily require “additional input":
>  - Similar to the features used in our experiment, they can be extracted from existing raw data (e.g., price and volume). In this case, the feature selection does not favor any particular sector and is broadly applicable to various assets.
>  - That said, in some cases, if an investor has hindsight regarding a range of assets (rather than individual ones), they could include non-asset-specific features (e.g., the QQQ index representing the tech sector) as part of the model. The incorporation of these features should follow configuration 2 (also as explained in Remark 3.3), alongside the generic asset-specific features.
> In our experiment with the sectors dataset, our model does not assign greater weight to the tech sector "XLK". The weight of "XLK" is, in fact, lower than that of "XLP" or “XLF” for most of the time. We have added an example asset allocation visualization (Figure 7, Appendix G.5, https://imgur.com/CCXyE7T) to demonstrate this.
>
> ## Choice Between Configurations
> >**Reviewer's Comment:** Can the authors provide a discussion on how to choose between the two proposed configurations?
>
> Yes, we have added a discussion explicitly on the choice of configurations. Below is the details:
>  - In the original paper version, we have discussed the distinctive characteristics of the two configurations in the paragraph after Remark 3.3 (page 5):
> > *“We showcase the feature-integrated Black-Litterman network as two configurations: one incorporating Effect 1 and another incorporating Effect 2. Intuitively, the first one better captures generic features while the second one more effectively handles the non-asset-related features.”*
>
>  - To make this concept more explicit to practice, we have added a statement after the above paragraph:
> > *“This implies that, in practice, if an investor takes generic features of assets (e.g. indicators derived from the time series of each asset, as shown in our experiment), configuration 1 should be used. If an investor takes features not specific to individual assets (e.g. interest rates), configuration 2 should be used. The two configurations are not contradicting, so one can take both types of features and incorporate them correspondingly.”*
>
> >**Reviewer's Comment:** in practice, practitioners primarily need a formula to compute portfolio weights, such a discussion would be valuable.
>
> Similar to the usage of the Black-Litterman (BL) Formula (Theorem 2.1), the practitioner formula to compute portfolio weights is provided by Theorem 3.2 and Theorem 3.3 corresponding to each configuration.
>
> >**Reviewer's Comment:** Similarly, what kinds of asset features are most suitable for the proposed method?
>
>
> Regarding what asset features are suitable, we do not limit the choice of data in this paper and only distinguish them for the purpose of choosing which configuration to use. For example, in our experiment, we randomly take 9 indicators of each asset price as generic features (see Appendix G.3 on page 25) to demonstrate the practice of configuration 1. In a features-driven model like ours, feature engineering literature [1][2] suggests taking diversified (uncorrelated) 	data to reduce multicollinearity so that it can provide robust mean/variance estimation and portfolio optimization decisions. However, detailed methods for feature selection problems in our model are not the focus of our work, thus we retain them for future works.
>
> [1]. Gujarati, Damodar, and Dawn Porter. "Multicollinearity: What happens if the regressors are correlated." Basic econometrics 363 (2003).
> [2]. Alin, Aylin. "Multicollinearity." Wiley interdisciplinary reviews: computational statistics 2, no. 3 (2010): 370-374.

---

### Official Review · Reviewer_f7Sa · 2025-03-13

**Overall Recommendation:** 3

**Summary:**

The paper presents a new formulation for the well-known Black-Litterman model, introducing a Bayesian reinterpretation of the model for portfolio optimization, eliminating the need for subjective investor views and their associated uncertainties. The authors analyse the problem from a theoretical perspective and numerically validate their findings.

---

Post-rebuttal: The authors provided adequate responses to my concerns. I will maintain my (positive) score.

**Claims And Evidence:**

All the claims are supported by evidence.

**Essential References Not Discussed:**

To the best of my knowledge, relevant literature is presented.

**Experimental Designs Or Analyses:**

The experimental validation is limited but coherent with the scope of the work.

**Methods And Evaluation Criteria:**

Yes, the evaluation criteria is coherent with the scope of the proposed model.

**Other Comments Or Suggestions:**

My only comment pertains to the highly specific target and scope of the work, which may make it less suitable for the general audience of ICML.

**Other Strengths And Weaknesses:**

I think this work is very well-presented, even if, due to space constraints, the authors often omitted discussions of their results and choices.

**Questions For Authors:**

No questions.

**Relation To Broader Scientific Literature:**

This work is relevant to the specific literature about this portfolio optimization model.

**Theoretical Claims:**

I didn't carefully check the proofs.

---

> ### Author Rebuttal · Authors · 2025-04-01
>
> Thanks for the reviews.
> The latest revision is readily available in the **[anonymous dropbox folder](https://www.dropbox.com/scl/fo/i13bhu138gjk76cf5r44v/ACog6LpDbYdBQbB87jtJA-Q?rlkey=mpv3b4xgbmr1cohaen6roxoyf&st=zndq6hnx&dl=0).**
> Any changes made from the submitted version are highlighted in blue in this updated draft.
>
> >**Reviewer's Comment:** My only comment pertains to the highly specific target and scope of the work, which may make it less suitable for the general audience of ICML.
>
> From a practical perspective, our work aims to take different types of data under different scenarios — some data is given, while others are not — to make predictions and decisions. Thus, it addresses machine learning problems using machine learning methods, and should be considered relevant to ICML community.
> Specifically, in our work,
> 1. **Different types of data** include: raw data, feature data extracted from raw data, feature data involving additional information, and heuristic expert knowledge (views). One of the major focuses in our model design is to capture the effects of these data through Bayesian networks with latent variables, a common methodology in machine learning research [1][2][3]. These data are clearly and consistently described in the introduction (page 1), Sec. 3.2, 3.3, and 3.4 (page 3-5).
> 2. **Different scenarios** refer to Sec. 3.3, where investor views are observed, and Sec. 3.4, where no subjective views are given. In the scenario of Sec. 3.4, our model is showcased by two configurations for handling different types of data. These scenarios are clearly and consistently described in the paragraph before Sec. 3.1 (page 3) and the paragraph after remark. 3.3 (page 5).
> 3. **Predictions** refer to the posterior probability distribution of unobserved asset returns given data estimated by each designed model. The prediction problems are described in problems 1, 2, 3 (page 2, 4, 5). The prediction models are Def 2.2, 3.3, 3.4, and 3.6, and they make predictions in Lemma 2.1, Corollary 3.1.1, 3.2.1, and 3.3.1, correspondingly.
> 4. **Decisions** refer to the solution of the portfolio optimization problem (Def. 2.1, page 2) based on the estimations of each designed model, including Lemma 2.1, Thm 3.1, 3.2, and 3.3. The portfolio optimization problem is also a popular topic in machine learning research [4][5][6][7][8].
> In this perspective, we deem our work to fall under the general scope of machine learning and satisfy the topics of interest (https://icml.cc/Conferences/2025/CallForPapers) of ICML.
>
> [1]. Anandkumar et al. (2013). Learning linear Bayesian networks with latent variables. ICML.
>
> [2]. Xie et al. (2016). Diversity-promoting Bayesian learning of latent variable models. ICML.
>
> [3]. Lorch et al. (2021). DiBS: Differentiable Bayesian structure learning. NeurIPS.
>
> [4]. Agarwal et al. (2006). Portfolio management via the Newton method. ICML.
>
> [5]. Qiu et al. (2015). Robust portfolio optimization. NeurIPS.
>
> [6]. Ito et al. (2018). Online portfolio selection with cardinality constraints: Regret bounds. NeurIPS.
>
> [7]. Tsai et al. (2023). Data-dependent bounds for online portfolio selection. NeurIPS.
>
> [8]. Lin et al. (2024). Globally optimal m-sparse Sharpe ratio portfolios. NeurIPS.
>
> >**Reviewer's Comment:** due to space constraints, the authors often omitted discussions of their results and choices.
>
> The general conclusion of this work is stated in Appendix A:
>
> *“We propose ...... real-world datasets (Section 4).”*
>
> We also have more detailed discussions for the results of each problem in Sec. 3:
>  - For Problem 2 (Sec 3.3), the discussion includes Remark 3.2 (classical Black-Litterman is a special case), Remark C.1/D.1 (the prediction has a ground truth limit), and a summary at the end of Sec. 3.3 (page 5).
>  - For Problem 3 (Sec 3.4), the discussion ofthe  SLP-BL model includes Remark 3.4 (equivalency to classical Black-Litterman) and a summary at the end of \textbf{Configuration 1} (page 6).
>  - For Problem 3 (Sec 3.4), the discussion of the FIV-BL model includes Remark C.2/D.2 (equivalency to SLP-BL model) after Thm. 3.3 and a summary at the end of \textbf{Configuration 2} (page 8).
>  - Due to the complex nature of configuration 2, we also discuss the further assumptions (conjugate prior) after Remark C.2 (page 7).
>
> To clarify the choice between configurations discussed in Section 3.4, we have added a guiding statement for practitioners at the end of page 5.
>
> *“This implies that, in practice, if an investor takes generic features of assets (e.g. indicators derived from the time series of each asset, as shown in our experiment), configuration 1 should be used. If an investor takes features not specific to individual assets (e.g. interest rates), configuration 2 should be used. The two configurations are not contradicting, so one can take both types of features and incorporate them correspondingly.”*

---

### Official Review · Reviewer_B3nT · 2025-03-16

**Overall Recommendation:** 4

**Summary:**

Paper removes the need for heuristic investor views while maintaining a Bayesian framework.
It makes the Black-Litterman model more data-driven, robust, and automated.

**Claims And Evidence:**

Claims are well supported.

**Essential References Not Discussed:**

All the essential referenes are well discussed.

**Experimental Designs Or Analyses:**

The backtests (2004-2024) are not enough (for the equities portfolio), and the small asset universe (38 assets) limits generalizability. As noted earlier, turnover, transaction costs, a longer backtest (1990s onward), and larger datasets (Asness et al. 2013, Fama-French 2015) should be included for robustness.

**Methods And Evaluation Criteria:**

The methods and evaluation criteria are reasonable, but the testing is limited to small asset universes (11-38 assets). To assess robustness, the model should be evaluated on larger datasets like those in Asness et al. (2013) and Fama-French (2015). Additionally, turnover analysis and net-of-transaction-cost performance are missing, which are crucial for real-world applicability.

Expanding the analysis to start at least in the 1990s and cover a much larger universe of assets would help assess the model’s robustness across more market regimes and economic conditions.

- Asness, C. S., Moskowitz, T. J., and Pedersen, L. H. (2013). Value and Momentum Everywhere. The Journal of Finance, 68(3):929–985​
- Fama, E. F. and French, K. R. (2015). A Five-Factor Asset Pricing Model. Journal of Financial Economics, 116(1):1–22​

**Other Comments Or Suggestions:**

No

**Other Strengths And Weaknesses:**

No

**Questions For Authors:**

No

**Relation To Broader Scientific Literature:**

The paper extends Bayesian Black-Litterman (Kolm & Ritter) by making views data-driven and explains its contributions well.

**Theoretical Claims:**

The proofs (Theorems 3.1, 3.2, 3.3) appear correct, following standard Bayesian inference.

---

> ### Author Rebuttal · Authors · 2025-04-01
>
> Thanks for the insightful questions and reviews.
> The latest revision is readily available in the **[anonymous dropbox folder](https://www.dropbox.com/scl/fo/i13bhu138gjk76cf5r44v/ACog6LpDbYdBQbB87jtJA-Q?rlkey=mpv3b4xgbmr1cohaen6roxoyf&st=zndq6hnx&dl=0).**
> Any changes or modifications made from the submitted version are highlighted in blue in this updated draft.
>
> >**Reviewer's Comment:** The backtests (2004-2024) are not enough (for the equities portfolio), and the small asset universe (38 assets) limits generalizability. As noted earlier, turnover, transaction costs, a longer backtest (1990s onward), and larger datasets (Asness et al. 2013, Fama-French 2015) should be included for robustness.
>
> In response to the suggestion of **backtesting periods**, we have added an extended version of our experiment on the Dow Jones Index to backtest from 1994 to 2024 (previously 2004-2024). As demonstrated in Table 2 (https://imgur.com/KcE5QxV), the general results remain consistent, with our model continuing to show (even greater) outperformance compared to the traditional Markowitz model. This outperformance is likely attributed to the more stable portfolio weights provided by our Bayesian-based model. We have added an example asset allocation visualization (Figure 7, Appendix G.5, https://imgur.com/CCXyE7T) to demonstrate this.
>
> Per the request for **turnover analysis**, we have added a new Appendix G.6 to compare the turnover rates between our SLP-BL model and the benchmark Markowitz model. The analysis includes Table 6 (https://imgur.com/Azl39Y0) presenting the average turnover rates, as well as visualizations of example turnover rates for both models (Figure 8 and 9, https://imgur.com/nqZ3rkk). Specifically, the average turnover rate for all the SLP-BL models is 24.79 on the DJI dataset and 19.51 on the sectors dataset, while for all the Markowitz models, the rates are 50.24 and 48.08, respectively.
>
> We would like to note that our experiments aim to prove the concept—specifically, to prove that this machine learning model could be readily implemented and useful (compared to the benchmarks):
>  - In the theoretical sections, we provide closed-form solutions for every model in our work. In experiments, we use Thm. 3.2 for the SLP-BL model, showing it is readily implemented.
>  - We take the asset sets, including the 41 DJI stocks and the 11 sectors, to represent the samples of at least large-cap equities. These sets are practically used as they provide some benefits compared to a larger set of stocks: ease of trading, smaller impact of slippage (due to higher liquidity), fewer trades per period (thus lower transaction costs), lower managerial costs, and lower computational costs.
>  - The monthly rebalance setting is also designed to enhance these benefits (particularly reduce the impact of transaction costs) while maintaining a similar performance compared to weekly or daily rebalance.
>
> Thus, in response to the reviewer’s suggestion regarding **asset universes** and **transaction costs**, we choose a smaller — but representative — set of assets for their practical usefulness, including the advantage of lower transaction costs. Our proof-of-concept experiment design also aims to minimize the effect of transaction costs, without explicitly considering them.
>
> Lastly, we need to clarify that the backtest period should be carefully chosen to avoid unfair comparison. Specifically, although the DJIA covers a long history, some stock data might be inaccessible. For example, companies like American Can Company, Navistar International Corporation, and USX Corporation were part of the DJIA before May 1991 but have since been delisted or replaced, making their data difficult to retrieve. Omitting these stocks (as many portfolio studies do) could introduce selection bias because delisted stocks, in the aftermath, often underperform. This is particularly concerning when backtesting over earlier years, as more data is unavailable. Notably, we observed better performance in the benchmark equal-weighted portfolio (EQW) but worse performance in other benchmarks - we suspect that the EQW may be benefiting from this selection bias. In terms of fair comparison, we believe our experimental results (20 yrs for sector ETFs; 30 yrs on the Dow Jones Index) are now sufficient for demonstrating the outperformance of our models.

---

### Decision · Program_Chairs · 2025-05-01

**Decision:**

Accept (poster)

**Comment:**

The authors propose what they describe as a Bayesian reformulation of the Black-Litterman model for portfolio optimization.

The reviewers acknowledged the relevance of the setting but initially raised several concerns regarding limited theoretical novelty, depth of the empirical validation, and inconsistencies between the paper's title and its actual scope. After the discussion, most of the concerns were addressed to a sufficient degree to move the needle to a positive score for all but one reviewer. Although this is a borderline submission, I believe it passes the threshold of acceptance, provided that the authors commit to incorporating all the points that were raised in the reviewing process in their revised version.